**Investigation of an extreme rainfall event during 8–12 December 2018 over central**
**Viet Nam – Part 2: An evaluation of predictability using a time-lagged cloud-resolving**
**ensemble system**
Chung-Chieh Wang[1], Duc Van Nguyen[1,2,*], Thang Van Vu[2], Pham Thi Thanh Nga[2], Pi-Yu
Chuang[1], and Kien Ba Truong[2,*]
Correspondence 1: kien.cbg@gmail.com
Correspondence 2: nguyenduc21e1@gmail.com
[1]Department of Earth Sciences, National Taiwan Normal University, Taipei, Taiwan
[2]Viet Nam Institute of Meteorology, Hydrology and Climate Change, Hanoi, Viet Nam
**Abstract:**
This is the second part of a two-part study that investigates an extreme rainfall event that
occurred from 8 to 12 December 2018 over central Viet Nam (referred to as the D18 event).
In this part, the study aims to evaluate the practical predictability of the D18 event using
the quantitative precipitation forecasts (QPFs) from a time-lagged cloud-resolving
ensemble system. To do this, 29 time-lagged (8 days in forecast range) high resolution (2.5
km) members were run, with the first member initialized at 12:00 UTC 3 December and
the last one at 12:00 UTC 10 December 2018. Between the first and the last members are
multiple members that were executed every 6 h. The evaluation results reveal that the
cloud-resolving model (CReSS) well predicted the rainfall fields at the short range (less
than 3 days) for 10 December (the rainiest day). Particularly, the CReSS shows high skill
in heavy-rainfall QPFs for this date with a Similarity Skill Score (SSS) greater than 0.5 for
both the last five members and the last nine members. The good results are due to the model
having good predictions of relevant meteorological variables such as surface winds.
However, the predictive skill is reduced at lead times longer than 3 days, and it is
challenging to achieve good QPFs for rainfall thresholds greater than 100 mm at lead times
longer than 6 days. These results also confirmed our scientific hypothesis that the cloud-

resolving time-lagged ensemble system (using the CReSS model) improved the QPFs of this event at the short range. Furthermore, the results also demonstrated that a decent QPF can be made at a longer lead time (by a member initialized at 1800 UTC 4 December).

In addition, the ensemble-based sensitivity analysis (ESA) of 24-h rainfall in central Viet Nam shows that it is highly sensitive to initial conditions, not only at lower levels but also at upper levels. The rainfall is sensitive to both kinematics and moisture convergence at low levels, and such sensitivities decrease with increasing lead time. The ESA also facilitates a better understanding of the mechanisms in the D18 event, implying that it is meaningful to apply ESA to control initial conditions in the future.

**1 Introduction**

The present study is the second part of a two-part study investigating the extreme rainfall event during 8–12 December 2018 over central Viet Nam (referred to as the D18 event hereafter). In this event, record-breaking rainfall occurred along the mid-central coast of Viet Nam, from Quang Binh to Quang Ngai provinces. The observation shows that the peak amount in rainfall accumulation, in particular, exceeded 800 mm over a 3-day period from 12:00 UTC 8 to 12:00 UTC 11 December (Fig. 1f). During this period, the rainiest day was 10 December with 24-h observed amount exceeding 600 mm at some stations (Fig. 4 OBS). This record-breaking rainfall event led to 13 deaths, widespread destructions in the environment and downstream cities, and heavy economic losses due to catastrophic flooding and landslides (Tuoi Tre news, 2018). In part 1 (Wang and Nguyen 2023), we focused on the analysis of the mechanism that caused this event and evaluated the simulation by the Cloud-Resolving Storm Simulator (CReSS; Tsuboki and Sakakibara, 2002, 2007). The analysis results point out the main factors which led to this event as well as its spatial rainfall distribution. These factors included the combined interaction between the strong northeasterly winds and easterly winds over the South China Sea (SCS) in the lower troposphere (below 700 hPa). The local terrain also played essential role due to its barrier effect. The cloud model's good simulation results in part 1 indicated its promising potential in forecasting this event. Hence, in part 2, the present study focuses on an

evaluation of its predictability of the D18 event through a series of time-lagged high-
resolution ensemble quantitative precipitation forecasts (QPFs) by the CReSS model.
Predicting heavy rainfall events is still challenging to meteorologists and weather
forecasters, although great progresses have been made in the science of numerical weather
prediction. The prediction of heavy to extreme rainfall is more difficult for Viet Nam,
where both multi-scale interactions among different weather systems and strong influence
by local topography often exist. For example, when D18 event occurred, several
operational models were unable to predict this event successfully. Specifically, Fig. 1
shows the predictions for the D18 event by three global models at the National Centers for
Environmental Prediction (NCEP), the European Centre for Medium-Range Weather
Forecasts (ECMWF), and the Japan Meteorological Agency (JMA), and by one mesoscale
regional model, the Weather Research and Forecasting (WRF) model, implemented for
operation at the Mid-central regional Hydro-Meteorological center in Da Nang city, Viet
Nam, with the finest horizontal grid spacing ($\Delta x$) of 6 km × 6 km. While these models
overall made good predictions in the surface wind field, their 72-h accumulated rainfall
amounts along the coast of central Viet Nam were less than 250 mm and much lower than
the observation, which exceeded 900 mm (Fig. 1). Therefore, in order to improve the QPFs
for heavy rainfall events in Viet Nam, we need to not only understand their mechanisms of
occurrence, but also adopt or develop better forecasting tools, more effective strategy, or
both.

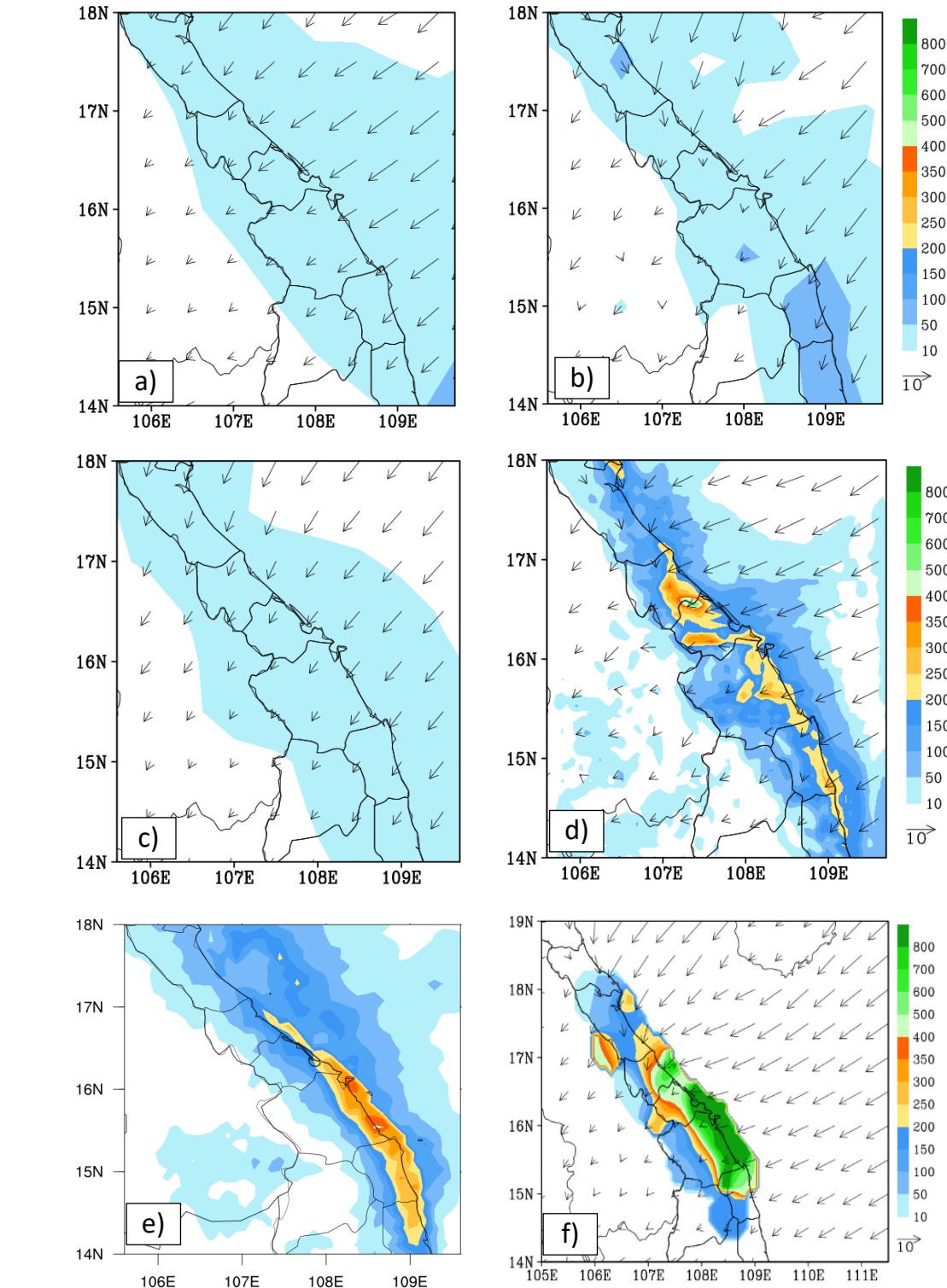

Figure 1. The predicted 72h accumulated rainfall (mm, shaded) and mean surface wind (ms$^{-1}$, vector) for the period of 12:00 UTC 8 December – 12:00 UTC 11 December 2018 obtained by (a) NCEP, (b) ECMWF, (c) JMA, (d) WRF, (e) 72h accumulated rainfall obtained by the Global Precipitation Measurement (GPM) estimate (IMERG Final Run

product) and (f) 72h in-situ observed accumulated rainfall (mm, shaded) and the mean surface wind derived from ERA5 data (ms$^{-1}$, vector), adapted from Fig. 14c of Wang and Nguyen (2023).

Among several different methods, present-day weather forecasts depend mainly on numerical weather prediction (NWP) using models, a scientific method that simulates weather and produce quantitative results (Fig. 1). However, there is always uncertainty in numerical forecasts due to the fact that the atmosphere is a chaotic system and tiny errors in the initial state can grow rapidly and lead to larger errors in the forecast (Hohenegger and Schär, 2007, Lorenz 1969). Various approximations in numerical methods are also sources of forecast uncertainty. Thus, by generating a range of possible weather conditions in days ahead or into the future, the ensemble forecasting was introduced as an effective method to estimate forecast uncertainty and improve the overall accuracy and usefulness of NWP products. This is because the ensemble mean typically has smaller errors than individual members, since the high predictability features that the members agree on are emphasized by the mean, while the low-predictability ones that the members do not agree on are filtered out or dampened (e.g., Leith 1974; Murphy 1988, Surcel et al. 2014). However, it may smooth out extreme events and underestimate their magnitude. Furthermore, some studies have shown high skill in QPFs for extreme rainfall produced by typhoons in Taiwan using the CReSS model, a cloud-resolving model (CRM), with high resolution and time-lagged approach (Wang et al. 2016; Wang 2015; Wang et al. 2014; Wang et al. 2013). Table 1 of Wang et al. (2016) shows that the high-resolution time-lagged ensemble forecasts provide overall better quality in comparison with both the traditional low-resolution ensemble forecasts and high-resolution deterministic forecasts at a comparable cost in computation.

Besides the advantages of ensemble forecasts described above, the ensemble-based sensitivity analysis (ESA) also provides an effective method to investigate how sensitive the forecast variables are and to what preceding factors. To be more specific, Torn and Hakim (2009) used ESA to evaluate how their subject, a group of tropical cyclones (TCs)

undergoing extratropical transition, in the prediction respond to changes in the initial condition. In their results, the cyclone minimum sea-level pressure forecasts are determined as strongly sensitive to TC intensity and position at short lead times and equally sensitive to mid-latitude troughs that interacted with the TC at longer lead times. For an extreme rainfall event in northern Taiwan, Wang et al. (2021) performed ESA using the results from 45 forecast members with grid sizes of 2.5–5 km to identify contributing factors to heavy rainfall. By normalizing their impacts on rainfall using standard deviation (SD), different factors can be compared quantitatively and on an equal footing. Ranked by their importance, these factors included the position of the surface Mei-yu front and its moving speed, the position of 700-hPa wind shift line and its speed, the moisture amount in the environment near the front, timing and location of frontal mesoscale low-pressure disturbance, and frontal intensity. Many other studies also used the ESA to study TCs, convective events, or support the development of operational ensemble sensitivity-based techniques to improve probabilistic forecasts (e.g., Kerr et al. 2019, Hu and Wu 2020, Coleman and Ancell 2020).

While ensemble-based sensitivity analysis provides valuable insights into key drivers of forecast outcomes as reviewed above, its effectiveness is inherently tied to the limits of predictability, which can vary by scale (Surcel et al. 2014, Surcel et al. 2015 and the references therein). Generally, the atmospheric predictability can be categorized into two types: practical predictability and intrinsic predictability (Melhauser and Zhang 2012, Nielsen and Schumacher 2016, Ying and Zhang 2017, Weyn and Durran 2018). Intrinsic predictability represents the highest achievable predictability using a nearly perfect initial conditions and a nearly perfect forecast model, and is mainly depended on scale and types of weather systems. Whereas, practical predictability describes the predictability using the best-available techniques and initial conditions, and therefore it can be limited by uncertainties in both the model and initial conditions. According to the studies cited above, practical predictability can be improved by improving the initial conditions, but it however

cannot exceed the intrinsic predictability (Ying and Zhang 2017). Based on these, in our
study, we investigate the practical predictability of the D18 event because it is a real event.
For heavy precipitation over central Viet Nam, Son and Tan (2009) used the Mesoscale
Model version 5 (MM5) to investigate the predictability of heavy-rainfall events over the
southern part of central Viet Nam during the period of 2005 and 2007. In this study,
experiments were configured for two nested domains with $\Delta x$ of 27 and 9 km, respectively.
Their results showed that the MM5 can predict heavy rainfall there and its performance is
better for events caused by TCs or TC interactions with the cold air. Toan et al. (2018)
assessed the predictability of heavy rainfall events in middle-central Viet Nam due to
combined effects of cold air and easterly winds using the WRF model within a forecast
range of 2 days. The model was also set with two nesting domains. The outer domain (D1)
covers the entire Vietnam and SCS with a $\Delta x$ of 18 km, while the inner domain (D2) focuses
on the Mid-Central Vietnam region with a $\Delta x$ of 6 km. The evaluation indicated that at 24-
h lead time, the model performed reasonably well at rainfall thresholds less than 100 mm
day$^{-1}$. At the 48-h forecast range, the model performed well only at thresholds below 50
mm day$^{-1}$ and had some skill at 50–100 mm day$^{-1}$. However, heavy-rainfall events at
thresholds over 100 mm day$^{-1}$ were almost unpredictable by the model.
Nhu et al. (2017) also used the WRF model to investigate the role of the topography in
central Viet Nam on the occurrence of a heavy-rainfall event there in November 1999. In
this study, the model with triply-nested domains with $\Delta x$ of 45, 15, and 5 km and 47 vertical
levels well simulated the northeast monsoon circulation, TCs, and the occurrence of heavy
rainfall in central Viet Nam. Furthermore, when the topography is removed, the three-day
total accumulated rainfall decreased sharply by approximately 75% compared to that in the
control experiment with the terrain.
Hoa Van Vo (2016) examined the predictability of heavy-rainfall events during the wet
seasons of 2008−2012 in the middle section and central highlands of Viet Nam using NWP
products from several global models, including the Global Forecasting System (GFS) of

NCEP, Global Spectral Model (GSM) of JMA, Navy Operational Global Atmospheric Processing System (NOGAPS) of the US Navy, and the Integrated Forecast System (IFS) of ECMWF. Their results indicated that the IFS and GSM performed better than the GFS and NOGAPS, and the IFS was evaluated the best. However, all four global models under-estimated rainfall in extreme events. One of the reasons for this under-estimation is that these models are global models, so their resolutions are too coarse for the relatively small study area.

The review above suggests that considerable limitations still exist in forecasting heavy rainfall in central Viet Nam, especially using coarser models. It also indicates that a high-resolution time-lagged ensemble approach may offer some advantages in the prediction of extreme rainfall events, such as a better simulation of local weather conditions, a quicker response to changes in forecast uncertainty in real time, and potentially a longer lead time for hazard preparation. Climatologically, the entire Viet Nam lies in the tropical zone (Fig. 2a), where vigorous but less organized convection often develops in response to local conditions. This region is also prone to the influence and interactions of weather systems spanning a wide range of scales as reviewed. In addition, although central Viet Nam is a small region with the narrowest place only about 80 km in width, it possesses significant topography running in the north-south direction to affect rainfall (Fig. 2a). Hence, a high-resolution CRM with detailed and explicit treatment in cloud microphysics is likely crucial for better QPFs for heavy rainfall in central Viet Nam.

Given the above review and analysis, the scientific hypotheses are proposed: Storm-scale processes and convection were important in the D18 event. However, both global and mesoscale models with a grid size down to 6 km × 6 km are not good enough for heavy-rainfall QPF without cloud-resolving capability (Fig. 1). Therefore, it is hypothesized that at higher resolution, the cloud-resolving time-lagged ensemble system (using the CReSS model) can improve the QPFs of this event at the short range. Additionally, this approach may also be able to extend the lead time of decent QPF beyond the short range. So, the goals of the study are to: 1) examine the hypothesis above, 2) investigate the (practical)

predictability of this event through a series of time-lagged ensemble predictions, including
whether a decent QPF can be made at a longer lead time, and 3) identify important factors
leading to this event, including the lead time of the signals of these factors, using the ESA
method. The rest of this paper is organized as follows. Section 2 describes the data, model,
and methodology used in the study. The model results are presented and evaluated in
Section 3. Finally, conclusions are offered in Section 4.
**2 Data and methodology**
2.1 Data
2.1.1 Model validation
*2.1.1.1 In-situ observation data*
The daily in-situ rainfall observations (12:00–12:00 UTC, i.e., 19:00–19:00 LST) from 8
to 12 December 2018 at 69 automated gauge stations across central Viet Nam are used for
case overview and verification of model results. This dataset is provided by the Mid-
Central Regional Hydro Meteorological Center, Viet Nam. The spatial distribution of these
gauge stations is depicted in Fig. 2b.
*2.1.1.2 The Global Precipitation Measurement (IMERG Final Run V07) data*
The Global Precipitation Measurement (GPM) is a joint international mission between the
National Aeronautics and Space Administration (NASA) and the Japan Aerospace
Exploration Agency (JAXA), employing a satellite network for advanced global rain and
snow observations. The GPM *IMERG Final Run* is a research-level product which is
created by intercalibrating, merging, and interpolating "all" satellite microwave
precipitation estimates along with microwave-calibrated infrared (IR) satellite estimates,
analyses from precipitation gauges, and potentially other precipitation estimation
methodologies at fine spatial and time scales. The horizontal resolution of this dataset is
$0.1° \times 0.1°$ latitude–longitude and the time interval is every 30 minute (Huffman et al.
2020). In this study, we used this satellite data (version 7) to verify rainfall distribution

over the coastal sea due to the limitation of the gauge network, where observations exist only inland as shown in Fig. 2b. The GPM IMERG data span from 12:00 UTC 8 to 12:00 UTC 11 December 2018 and are used to analyze the D18 event as well as the rainiest day of this event (10 December).

*2.1.1.3 The NCEP GDAS/FNL global tropospheric analyses data*

The present study used this dataset (version d083003) to verify initial data and model outputs. The NCEP FNL analysis is an operational global gridded analysis and is freely provided by the NCEP. The horizontal resolution of this dataset is $0.25° \times 0.25°$ latitude–longitude with 26 levels extending from the surface to 10 hPa. The temporal interval is 6 h. The variables used in this study include the zonal and meridional wind components, relative humidity, and vertical velocity at 925 hPa covering the case period from 18:00 UTC 4 to 12:00 UTC 9 December 2018.

2.1.2 The added values of CReSS ensemble

*2.1.2.1 The International Grand Global Ensemble retrieval*

In this study, we used the global model predictions to analyze the predictability of the D18 event. The International Grand Global Ensemble (TIGGE) is a key component of The Observing System Research and Predictability Experiment (THORPEX) research program, whose aim is to accelerate the improvements in the accuracy of 1-day to 2-week high-impact weather forecasts. The TIGGE provides not only deterministic forecast data but also ensemble prediction datasets from major centers, including NCEP of the USA, ECMWF of the European countries, and JMA of Japan, since 2006. This dataset has been used for a wide range of research studies on predictability and dynamical processes. the variables utilized included total precipitation and surface winds (at 10-m height) from NCEP, ECMWF, and JMA at 6-h intervals during the data period from 12:00 UTC 8 to 12:00 UTC 11 December 2018 (as shown in Figs. 1a-c). The link to this dataset is placed in the "code and data availability" section.

2.2.1 Model description and experiment setup

We used the Cloud-resolving Storm Simulator (CReSS) developed by Nagoya University,
Japan (Tsuboki and Sakakibara, 2002, 2007). This is a non-hydrostatic and compressible
cloud model, designed for simulation of various weather events at high (cloud-resolving)
resolution. In the model, the cloud microphysics is treated explicitly at the user-selected
degree of complexity, such as the bulk cold-rain scheme with six species: vapor, cloud
water, cloud ice, rain, snow, and graupel (Lin et al., 1983; Cotton et al., 1986; Murakami,
1990, 1994; Ikawa and Saito, 1991). Other subgrid-scale processes parameterized, such as
turbulent mixing in the planetary boundary layer and physical options for surface
processes, including momentum/energy fluxes, shortwave and longwave radiation, are
summarized in Table 1.
For the initial and boundary conditions (IC/BCs), the NCEP GFS analyses and
deterministic forecast runs, executed every 6 h at 00:00, 06:00, 12:00, and 18:00 UTC daily
(dataset ds084.6), were used to drive the CReSS model predictions. The horizontal
resolution of the data is $0.25° \times 0.25°$, and 26 of vertical levels, and the forecast fields are
provided every 3 h from the initial time out to a range of 192 h. The data link is also placed
in the "code and data availability" section.
To evaluate of the predictability of the D18 event using an ensemble time-lagged high-
resolution system and investigate the ensemble sensitivity of variables for the rainfall, 29
experiments were performed. The first member was initialized at 12:00 UTC on 3
December and the last one at 12:00 UTC on 10 December 2018. Between them, a new
member was initialized every 6 h and all members have a simulation length of 192 h. All
experiments used a single domain at 2.5 km horizontal grid spacing and a dimension in ($x$,
$y$, $z$) of $912 \times 900 \times 60$ grid points (Table 1, cf. Fig. 2). As mentioned above, the NCEP
GFS was used as the IC/BCs of the CReSS model.

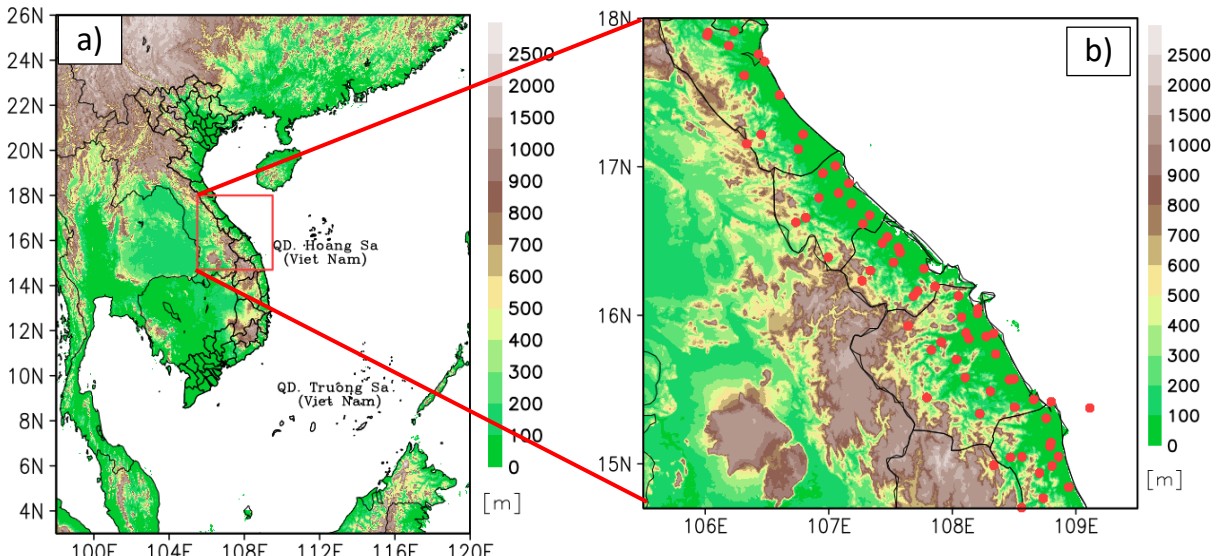


**Figure 2**. (a) The simulation domain of the CReSS model and topography (m, shaded) used in the study. The red box marks the study area. (b) The distribution of the observation stations (red dots) in the study area.


Table 1. The basic information of experiments.

| Domain and Basic setup | |
| --- | --- |
| Model domain | 3°–26°N; 98°–120°E |
| Grid dimension (x, y, z) | 912 × 900 × 60 |
| Grid spacing (x, y, z) | 2.5 km × 2.5 km × 0.5 km* |
| Projection | Mercator |
| IC/BCs (including SST) | NCEP GDAS/FNL Global Gridded Analyses and Forecasts (0.25° × 0.25°, every 6 h, 26 pressure levels) |

| | |
|---|---|
| Topography (for CTRL only) | Digital elevation model by JMA at (1/120)° spatial resolution |
| Simulation length | 192 h |
| Output frequency | 1 hour |
| **Model physical setup** | |
| Cloud microphysics | Double-moment Bulk cold-rain scheme (six species, Lin et al., 1983; Cotton et al., 1986; Murakami, 1990, 1994; Ikawa and Saito, 1991) |
| PBL parameterization | 1.5-order closure with prediction of turbulent kinetic energy (Deardorff, 1980; Tsuboki and Sakakibara, 2007) |
| Surface processes | Energy and momentum fluxes, shortwave and longwave radiation (Kondo, 1976; Louis et al., 1982; Segami et al., 1989) |
| Soil model | 41 levels, every 5 cm deep to 2 m |

* The vertical grid spacing ($\Delta z$) of CReSS is stretched (smallest at bottom) and the
averaged value is given in the parentheses
2.3 Verification of model rainfall
In order to verify model-simulated rainfall, some verification methods are used, including
(1) visual comparison between the model and the observation (from the 69 automated
gauges over the study area), and (2) objective verification using categorical skill scores at
various rainfall thresholds from the lowest at 0.05 mm up to 900 mm for three-day total.
These scores are presented below along with their formulas and interpretation. To apply
these scores at a given threshold, the model and observed value pairs at all verification
points $N$ (gauge sites here) are first compared and classified to construct a $2 \times 2$ contingency
table (Wilks, 2006). At any given site, if the event takes place (reaching the threshold) in
both model and observation, the prediction is considered a hit ($H$). If the event occurs only
in observation but not the model, it is a miss ($M$). If the event is predicted in the model but
not observed, it is a false alarm ($FA$). Finally, if both model and observation show no event,
the outcome is correct rejection ($CR$). After all the points are classified into the above four
categories, the categorical scores can be calculated by their corresponding formula as:
Bias Score (BS) $= (H + FA)/(H + M)$,          (1)
Probability of Detection (POD) $= H/(H + M)$,          (2)
False Alarms Ratio (FAR) $= FA/(H + FA)$,          (3)
Threat Score (TS) $= H/(H + M + FA)$.          (4)
The values of TS, POD, and FAR are all ranged from 0 to 1, and the higher the better for
both TS and POD, but the opposite for FAR. For BS, its possible value can vary from 0 to
$N$ and indicate overestimation (underestimation) by the model for the events if greater than
(less than) unity.
2.3.1 The Similarity Skill Score
In addition to the categorical scores, the Similarity Skill Score (SSS, Wang et al., 2022) is
also applied to evaluate the model rainfall results, as

$$\text{SSS} = 1 - \frac{\frac{1}{N}\sum_{i=1}^{N}(F_i - O_i)^2}{\frac{1}{N}\sum_{i=1}^{N}F_i^2 + \frac{1}{N}\sum_{i=1}^{N}O_i^2},\qquad (5)$$


where $N$ is the total number of verification points as before, and $F_i$ is the forecast rainfall
amount and $O_i$ is the observed value at the $i$th point among $N$, respectively. The SSS is a
measure against the worst mean squared error (MSE) possible. The formula shows that a
forecast with perfect skill has an SSS of 1, while a score of 0 means zero skill when the
model rainfall does not overlap with the observation anywhere.
Note that even though Eq. (5) has the same form as the Fractions Skill Score (Roberts and
Lean, 2008), the SSS is not a neighborhood method. Thus, it is suited for QPF verifications
where the rainfall location is important (as in our case).
2.3.2. The ensemble spread (standard deviation)
The ensemble spread is a measure of the difference among the members about the ensemble
mean, and one suitable parameter is the standard deviation (SD). In other words, the
ensemble spread reflects the diversity of all possible outcomes. Hence, the ensemble spread
is often applied to describe the magnitude of the forecast errors. For a well calibrated
ensemble, for example, a small spread indicates high theoretical forecast accuracy (and low
uncertainty), and vice versa for a large spread (Cattoën et al. 2020). Using the SD, the
spread is computed by the formula below:

$$\text{SD} = \sqrt{\frac{\sum_{i=1}^{n}(x_i - \mu_x)^2}{n-1}} \quad , \quad\quad\quad (6)$$


where $x_i$ is the predicted value of member $i$ for the variable $x$, $\mu_x$ is the ensemble mean,
and $n$ is the total number of ensemble members, respectively.
2.3.3. Ensemble Sensitivity Analysis
As mentioned above, an ensemble forecast is a set of forecasts produced by many separate
forecasts typically with different initial conditions. Moreover, as we know, NWP outcomes
are often sensitive to small changes in ICs and the sensitivity analysis is considered a
method to improve forecasts through targeting observations. Hence, this study used the
ESA method introduced by Ancell and Hakim (2007) to examine how a forecast variable
responds to changes in ICs. The ensemble sensitivity is computed by the formula:

$$\frac{\partial R}{\partial x_t} = \frac{COV(R, x_t)}{VAR(x_t)} . \quad\quad\quad (7)$$


Here, the response function $R$ is chosen to be the areal-mean 24-h accumulated rainfall in
central Viet Nam (15.5°-16.3°N, 107.9°-108.6°E) on the rainiest day, from 12:00 UTC 9
to 12:00 UTC 10 December 2018. The starting time of this period, i.e., 12:00 UTC 9
December, is defined as $t_0$. Various scalar variables are considered for $x_t$, at a time from 48
h earlier ($t_{-48}$, or 12:00 UTC 7 December) to $t_0$ at 24-h intervals. The *COV* is the covariance
of $R$ and $x_t$, and *VAR* is the variance of $x_t$, respectively.
Since the analysis in part 1 has identified that the D18 event was caused by the combined
effect between the atmospheric disturbances at lower levels, such as the cold surge and
easterly wind, and the topography, the ESA herein has been applied to selected variables
at surface, near-surface, and mid-tropospheric levels to assess the sensitivity of the rainfall
field to ICs and its predictability. In order to facilitate the comparison among the impacts
of different variables, this study normalized ESA results by using the standardized anomaly
in the denominator of Eq. (7) and expressed them as the change in $R$ (in mm) in response
to an increase in $x_t$ by one SD in subsequent sections.
**3 Model results**
3.1 Time-lagged 24-h QPFs by the CReSS model
In this section, time-lagged forecasts targeted for the 24-h period from 12:00 UTC on 9 to
12:00 UTC on 10 December in the D18 event by the 2.5-km CReSS model are presented
and evaluated. This 24-h period is chosen because it is the rainiest day with in-situ
observation exceeding 600 mm at some stations (Fig. 3 OBS). Figure 3 shows 25 possible
scenarios of 24-h rainfall and average surface winds over the target period produced by the
lagged runs every 6 h, with the earliest initial time at 12:00 UTC 3 December and the latest
one at 12:00 UTC 9 December 2018. It is immediately clear that several members made a
rather good 24-h QPF not only in amounts, but also in rainfall location and spatial
distribution. These include most members starting during 8-9 December, and also an
impressive member from 18:00 UTC 4 December. In this latter run, a reasonably good QPF
was produced at a rather long lead time, almost five days (114 h) prior to the beginning of
the target period. A common feature among these good members is that they all captured
the direction and magnitude of surface winds quite well. On the other hand, most other

members were less ideal in their QPFs when initialized before 06:00 UTC on 7 December at lead times beyond two days (before the target period). In general, they also did not predict the surface winds well enough.


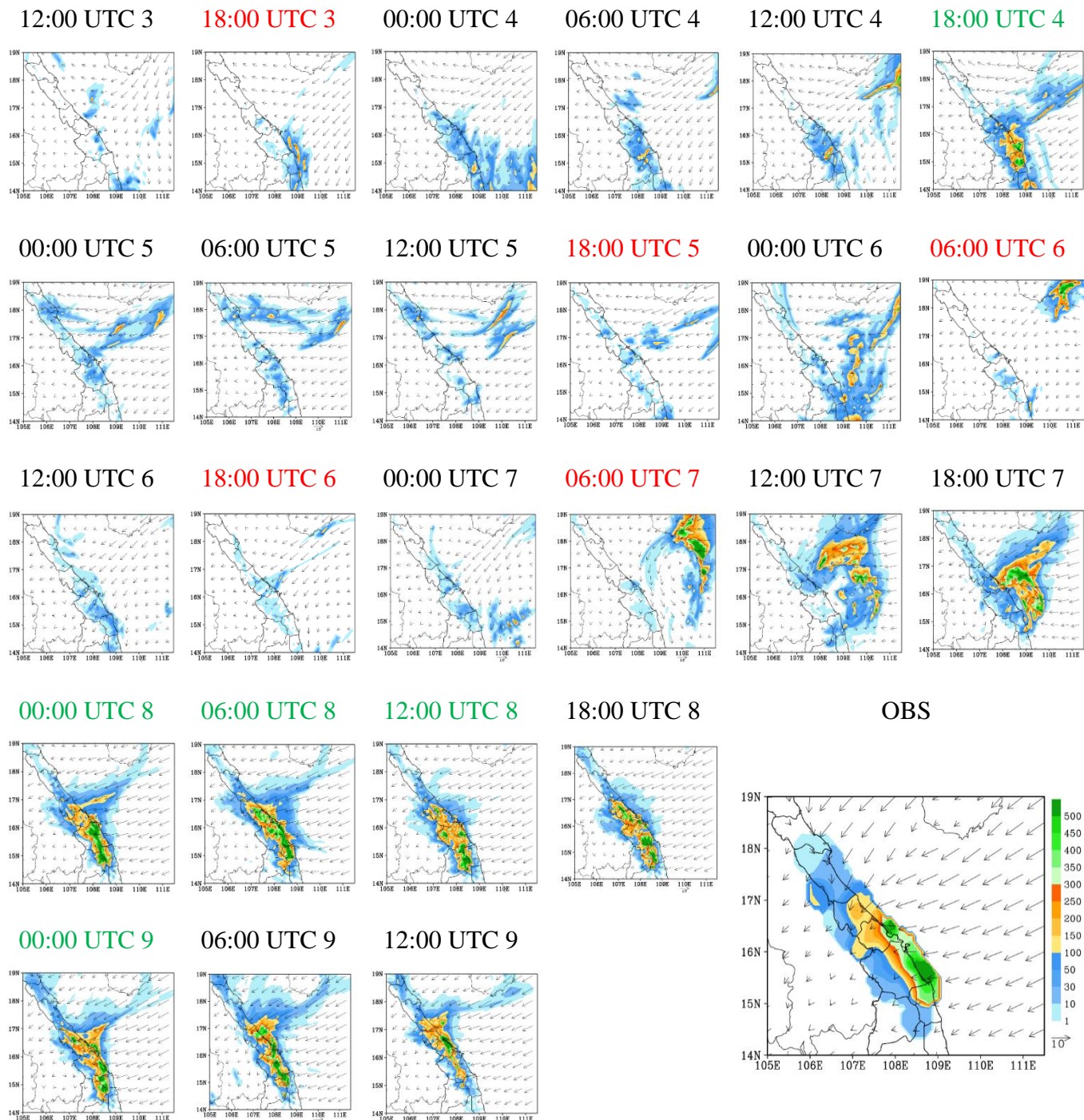

**Figure 3**. The predicted 24h accumulated rainfall (mm, shaded, scale on the right of panel OBS) and the mean surface horizontal wind (ms$^{-1}$, vector, reference length at panel OBS) on 10 December 2018 (from 12:00 UTC 9 December to 12:00 UTC 10 December 2018). The green color mark good members and the red color marks bad members. In OBS, 24h in-situ observed rainfall (mm, shaded) and the surface wind derived from ERA5 data (ms$^{-1}$, vector), adapted from Fig. 12f of Wang and Nguyen (2023).

Furthermore, as we know, ensemble weather forecasts are a set of forecasts from multiple members that represent the range of future weather possibilities, and the simplest way to use them is through the ensemble mean, which emphasizes the features that the members agree upon. In order to see how well the 2.5-km CReSS can predict the D18 event with the time-lagged strategy in terms possible scenarios of 24-h accumulated rainfall for 10 December, lagged runs are grouped based on their range of initial times in Fig. 4. It can be clearly seen that the rainfall predictions by the fifth four (executed between 12:00 UTC 7 and 06:00 UTC 8 December) and the sixth four members (between 12:00 UTC 8 and 06:00 UTC 9 December) are quite similar to the observation, not only in rainfall amount but also in the locations of concentrated rainfall. For other subgroups, the rainfall was much lower than the observation in their scenarios. In which, the rainfall accumulations from the third (12:00 UTC 5 to 06:00 UTC 6 December) and fourth four (12:00 UTC 6 to 06:00 UTC 7 December) members are the lowest. One relevant assessment to the outcome of these eight runs is that none of them predicted the surface wind field well enough at their ranges (beyond three days), as discussed previously. On the other hand, the mean rainfall from the second four members (12:00 UTC 4 to 06:00 UTC 5 December) is the best among all subgroups at the extended range due to a single good forecast initialized at 18:00 UTC on 4 December [cf. Fig. 3 (18:00 UTC 4)].

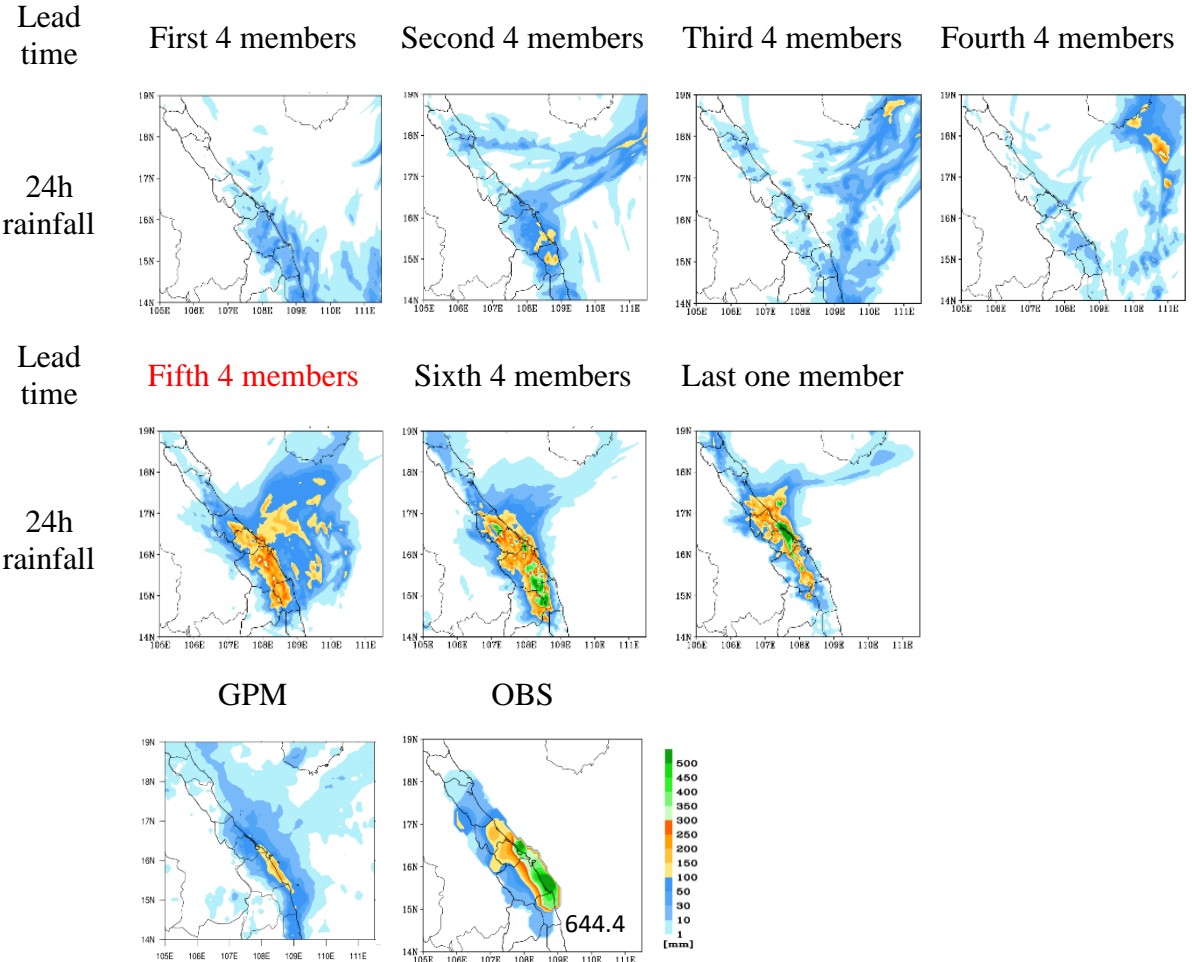

**Figure 4**. The predicted 24h rainfall by subgroup members, 24h accumulated rainfall by the Global Precipitation Measurement (GPM) estimate (IMERG Final Run product), 24h observed rainfall (mm, peak amount labeled at the lower-right corner) for the period of 12:00 UTC 9 December – 12:00 UTC 10 December 2018 as labelled. The same color bar (lower right) is used for all panels.

Besides the evaluation on time-lagged results using batches of successive runs (every 4 members) as presented above, this study also grouped the members using different ensemble sizes based on their behavior in order to better assess the temporal evolution of forecast uncertainty and event predictability as the lead time shortened. Particularly, the 25 members were divided into several subgroups as shown in Fig. 5, including the first eight

members (those executed during 12:00 UTC 3−06:00 UTC 5 December), the middle eight
members (runs between 12:00 UTC 5 and 06:00 UTC 7 December), the last nine members
(12:00 UTC 7−12:00 UTC 9 December), and the last five members (12:00 UTC 8−12:00
UTC 9 December), respectively. In other words, the last five members were those executed
within 24 h (1 day) prior to the beginning of the target period, and so on.
In Fig. 5, it is clear that both the ensemble means from the last five and the last nine
members compare quite favorably to the observation, not only in the accumulated amount
but also in spatial distribution of rainfall. This indicates that the model could produce QPFs
at fairly good quality and rather consistently since the time as early as roughly 48 h prior
to the commencement of the rainfall event (also Fig. 3). These two sub-groups within the
short range gave much better quality in QPFs than the other sub-groups executed before
them at longer lead times, including the first eight, middle eight, and all 25 members.
In terms of skill scores, for example, the mean QPF by the last five members have TS =
0.4, POD = 0.8, BS = 1.5, and FAR = 0.5 at 100 mm (per 24 h), while the last nine members
give similar scores of TS = 0.5, POD = 0.8, BS = 1.4, and FAR = 0.5 (Figs. 6a-d),
respectively. On the contrary, the mean QPFs from both the first and middle eight members
only yield zero scores in TS, POD, and BS with no skill in FAR at 100 mm (and above),
obviously due to not enough rainfall in central Viet Nam in most of their members. At 200
mm (per 24 h), similarly, the last five members (TS = 0.2, POD = 0.4, BS = 1.4, and FAR
= 0.7) and the last nine members (TS = 0.3, POD = 0.5, BS = 1.2, and FAR = 0.6) again
produce much better scores in QPFs, compared to no skill in all four scores in QPFs from
the middle eight, first eight, and all 25 members (Figs. 6a-d). In SSS, the mean from the
last nine members exhibits the highest score (0.64), the middle eight members have the
lowest score (0.04), and the mean from all 25 members is 0.43 (Fig. 6e).

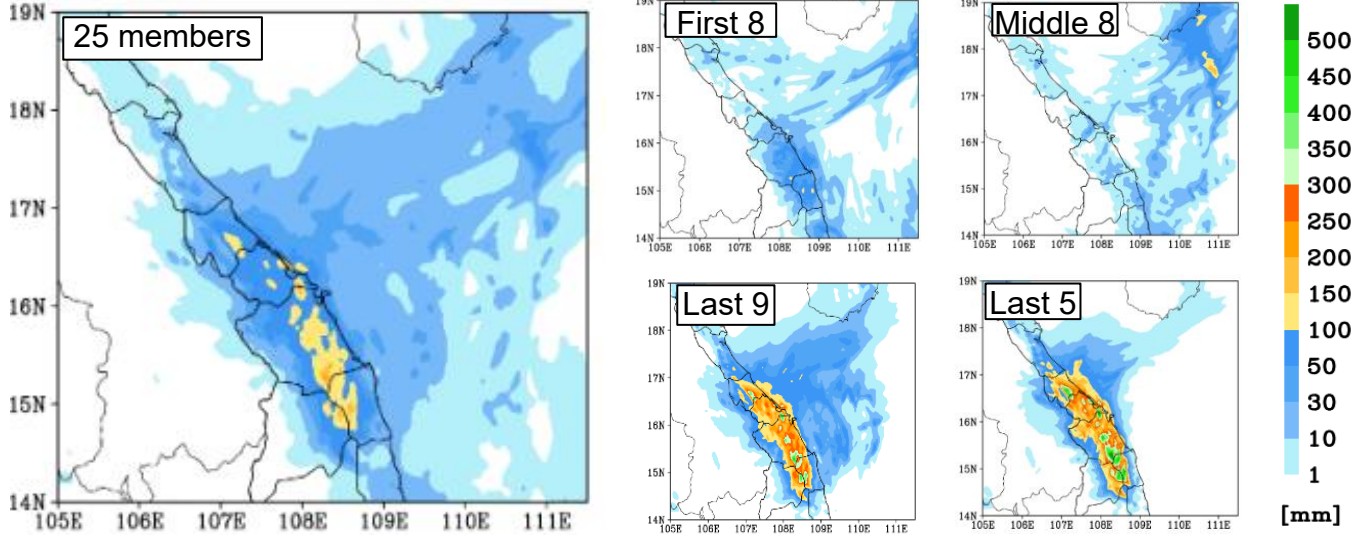


**Figure 5**. Ensemble mean rainfall (shaded, scale on the right) from all 25 time-lagged members, executed every 6 h from 12:00 UTC 3 December to 12:00 UTC 9 December, for the 24h period from 12:00 UTC 9 December to 12:00 UTC 10 December.

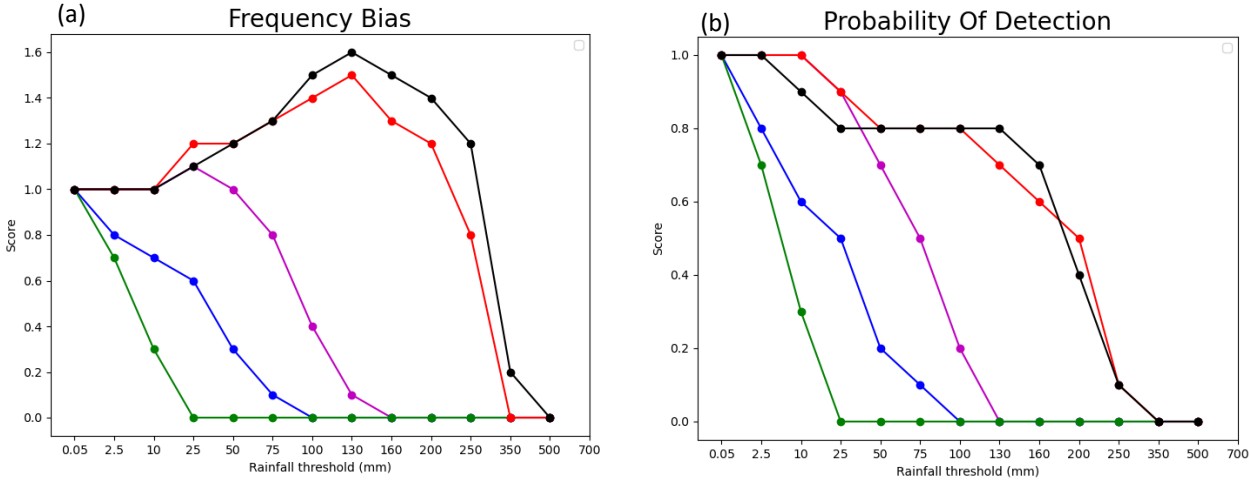

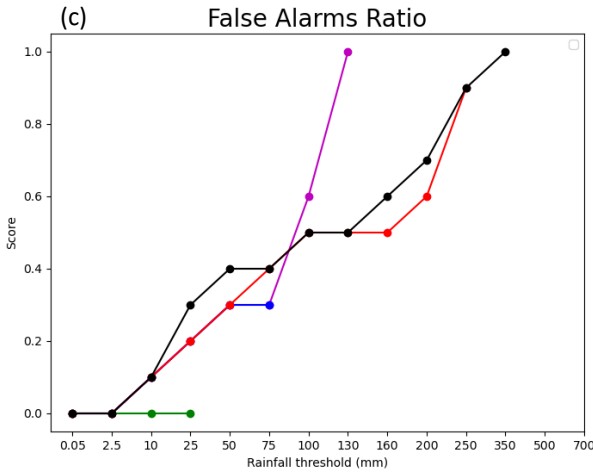

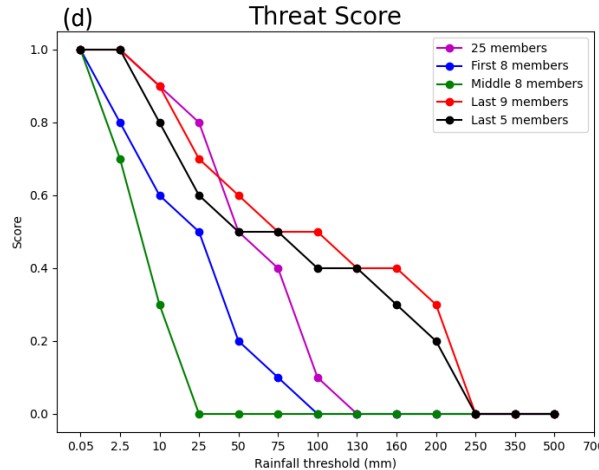

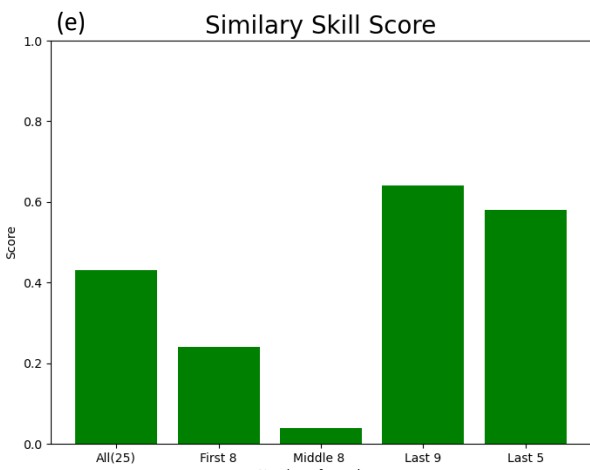

**Figure 6**. Statistic scores for 24h mean rainfall, obtained from twenty-five 8-day forecasts for 10 December 2018 [from 12:00 UTC 9 December to 12:00 UTC 10 December].


However, as indicated by the SD, the spreads in rainfall scenarios in both ensembles from the last five and nine members are quite large (Fig. 7). Thus, while the lagged members can produce a wide range of possible rainfall scenarios for the D18 event, which is the main purpose of an ensemble as reviewed in Section 1, the members often cannot agree on the precise locations of heavy rainfall. Given the small scale of local convection during the event, this result is perhaps anticipated. On the other hand, the maxima in spread are >160 mm in Fig. 7 among the last nine members, perhaps quite reasonable in magnitude compared to the peak amounts of about 400 mm in the ensemble mean. In any case, Figs. 6 and 7 indicate that the predictability of the D18 event changed considerably with time,

and the 2.5-km CReSS has a good skill in QPFs inside the short range (≤ 72 h). However,
it remains difficult to predict the event successfully at longer lead times.

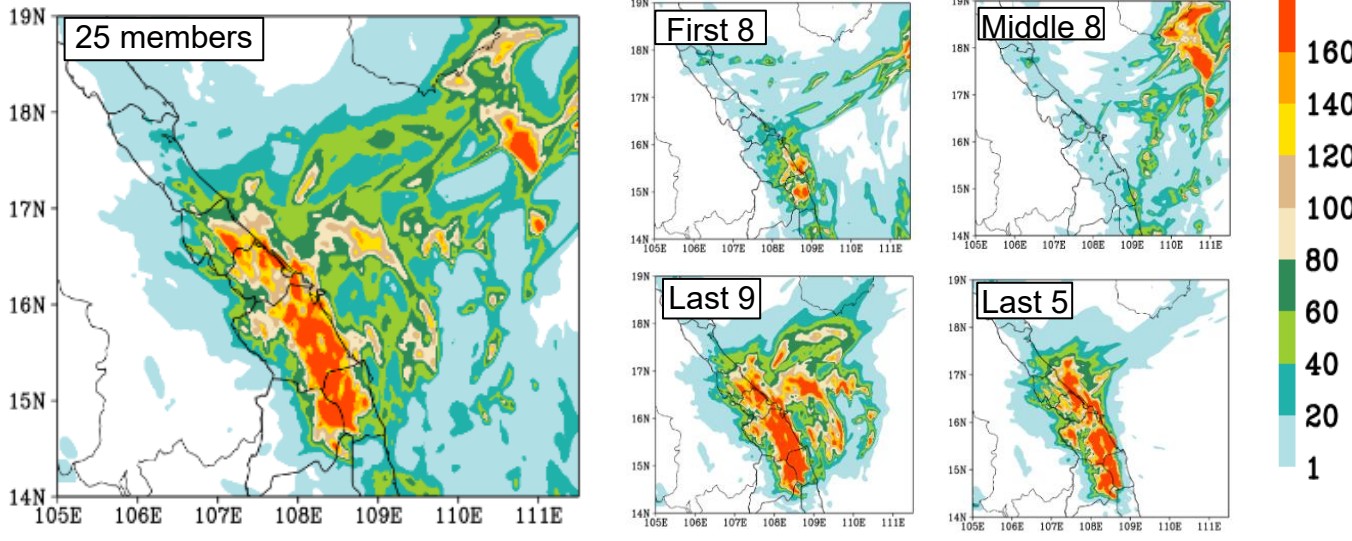


**Figure 7**. The spread (shaded, scale on the right) from all 25 time-lagged members,
executed every 6 h from 12:00 UTC 3 December to 12:00 UTC 9 December, for the 24h
period from 12:00 UTC 9 December to 12:00 UTC 10 December.

The probability information derived from the sub-ensemble groups at four different rainfall
thresholds from 100 to 450 mm is shown in Fig. 8, in which the increase in heavy-rainfall
probability in central Viet Nam and thus the predictability of the event with time is also
evident. From the first eight members executed at the longest range (≥ 102 h prior to rainfall
accumulation), there is only a 10-25% chance in parts of central Viet Nam to receive at
least 100 mm of rainfall for 10 December (from 12:00 UTC 9 to 12:00 UTC 10 December).
The probability is even lower from the middle eight members (run between 54-96 h prior
to target period), as their SSS is the lowest among all sub-ensemble groups and only a
couple of the runs could reach 100 mm anywhere inland in central Viet Nam. As the lead

time shortens to inside the short range, the probabilities to have ≥ 100 mm of rainfall increase dramatically, to roughly 70-80 % in the last nine members and further to over 80-90% in the last five members. Due to the contribution from later members, about 20-40% of all 25 members can reach 100 mm inland. Toward higher thresholds, the probabilities decrease in Fig. 8 as expected, so do the areal sizes actually reaching those thresholds (pink contours). At the highest value of 450 mm, the ensembles in general show less than about 20%-30% chance for its occurrence from the last five and last nine members, respectively, and the high probability areas are also slightly more inland than the observed one.

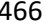

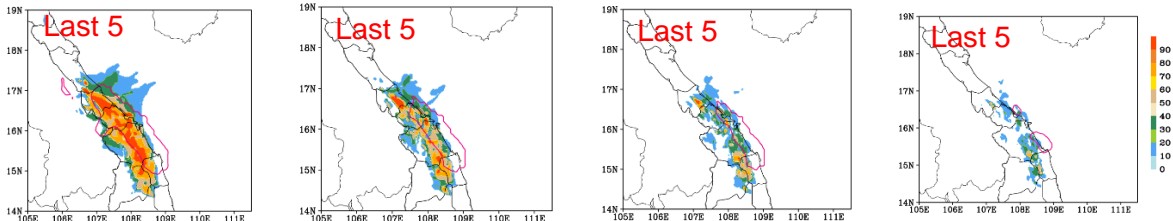

**Figure 8**. Probability distribution (%; shaded, scale on the right) from all 25 time-lagged members, executed every 6 h from 12:00 UTC 3 December to 12:00 UTC 9 December, reaching thresholds of 100, 200, 300, and 450 mm, for the 24h period from 12:00 UTC 9 December to 12:00 UTC 10 December. The observed areas at the same thresholds are depicted by the pink contours. For each picture, red labeled at the top-left corner show the number of members grouped to calculate the probability distribution.

3.2 Ensemble-based sensitivity analysis

The results in Section 3.1 above reveal that the CReSS model with a horizontal grid size of 2.5 km predicted good QPFs for the rainiest day of the event and performed better than those reviewed in Section 1. Therefore, relying on this good performance, the ESA is carried out in this subsection.

Firstly, five good members (those with initial times at 18:00 UTC on 4, 00:00, 06:00, and 12:00 UTC on 8, and 00:00 UTC on 9 December) and five bad ones (those ran at 18:00 UTC on 3, 18:00 UTC on 5, 06:00 and 18:00 UTC on 6, and 06:00 UTC on 7 December) are chosen and by using their differences (good minus bad members), Fig. 9 shows that the main reason for the significantly different forecast outcomes lies in differences in the input datasets (i.e., IC/BCs). Specifically, the surface easterly winds were much stronger and the relative humidity much higher surrounding central Viet Nam and its upstream areas in the GFS forecast data valid at 12:00 UTC on 9 December (used as BCs in CReSS runs) in the good members than in the bad ones (Fig. 9a). Subsequently, the good CReSS members produced much more rainfall in central Viet Nam (Fig. 9b). These factors were also identified as crucial for the extreme rainfall in the D18 event in Part 1.

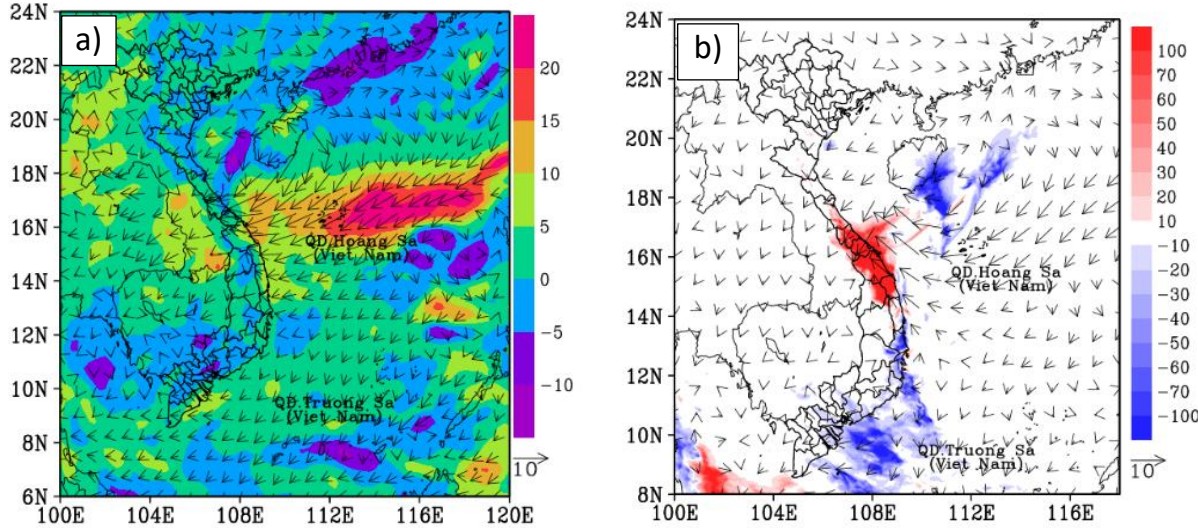

489

**Figure 9**. The difference in (a) input data (boundary conditions) and (b) CReSS output between averaged 5 good members (members ran at 18:00 UTC 4, 00:00 UTC 8, 06:00 UTC 8, 12:00 UTC 8, 00:00 UTC 9) and 5 bad members (members ran at 18:00 UTC 3, 18:00 UTC 5, 06:00 UTC 6, 18:00 UTC 6, 06:00 UTC 7). For input data, relative humidity (%, shaded) and surface wind (ms$^{-1}$, vector) at 12:00 UTC December 9 2018. For CReSS output, 24-h accumulated rainfall (mm, shaded) and surface wind (ms$^{-1}$, vector).

Meanwhile, Fig. 10 shows the difference in the evolution of synoptic-scale patterns (features) zoomed into the study area. To be more specific, Fig. 10a depicts the difference (CReSS output minus NCEP FNL analysis) in the horizontal wind and vertical velocity between the averages of the 5 good members and the NCEP FNL analysis at 925 hPa at 12:00 UTC 9, and it is small although each member was initialized at a different lead time. It implies that these members captured well the evolution of weather patterns of this event. Additionally, the model vertical velocity is seen to be stronger than the NCEP FNL data. Therefore, these members produced the rainfall closer to the observation with the presence of complex terrain in the study area. On the contrary, bad members did not capture the evolution of weather patterns well enough (Fig. 10b), and they could not produce good QPFs as a result.

Furthermore, Fig. 10d indicates very small differences in the IC of the member that was
initialized at 18:00 UTC 4 to the FNL analysis (thus suggesting smaller errors), especially
over the study area. From this initial data, the evolution of weather patterns in this CReSS
run also agreed well with the analyses during the first three days (not shown), and the
differences remained relatively small even at 12:00 UTC 9, at a lead time of roughly 5 days
(Figs. 10e,f,g). Compare to this, a bad member initialized at 18:00 UTC 6 (at a shorter lead
time by 2 days) exhibited somewhat larger differences in the initial state in relation to the
NCEP FNL analysis (Fig. 10h). This difference then led to larger and more evident
differences in weather patterns, as seen in Figs. 10 i, j, k by this particular member that
performed worse in QPFs (member ran at 18 UTC 06). The results here not only indicate
that it is still possible to have good rainfall forecasts at a lead time up to 5 days, but also
show some predictability by a cloud-resolving model at such long lead times.



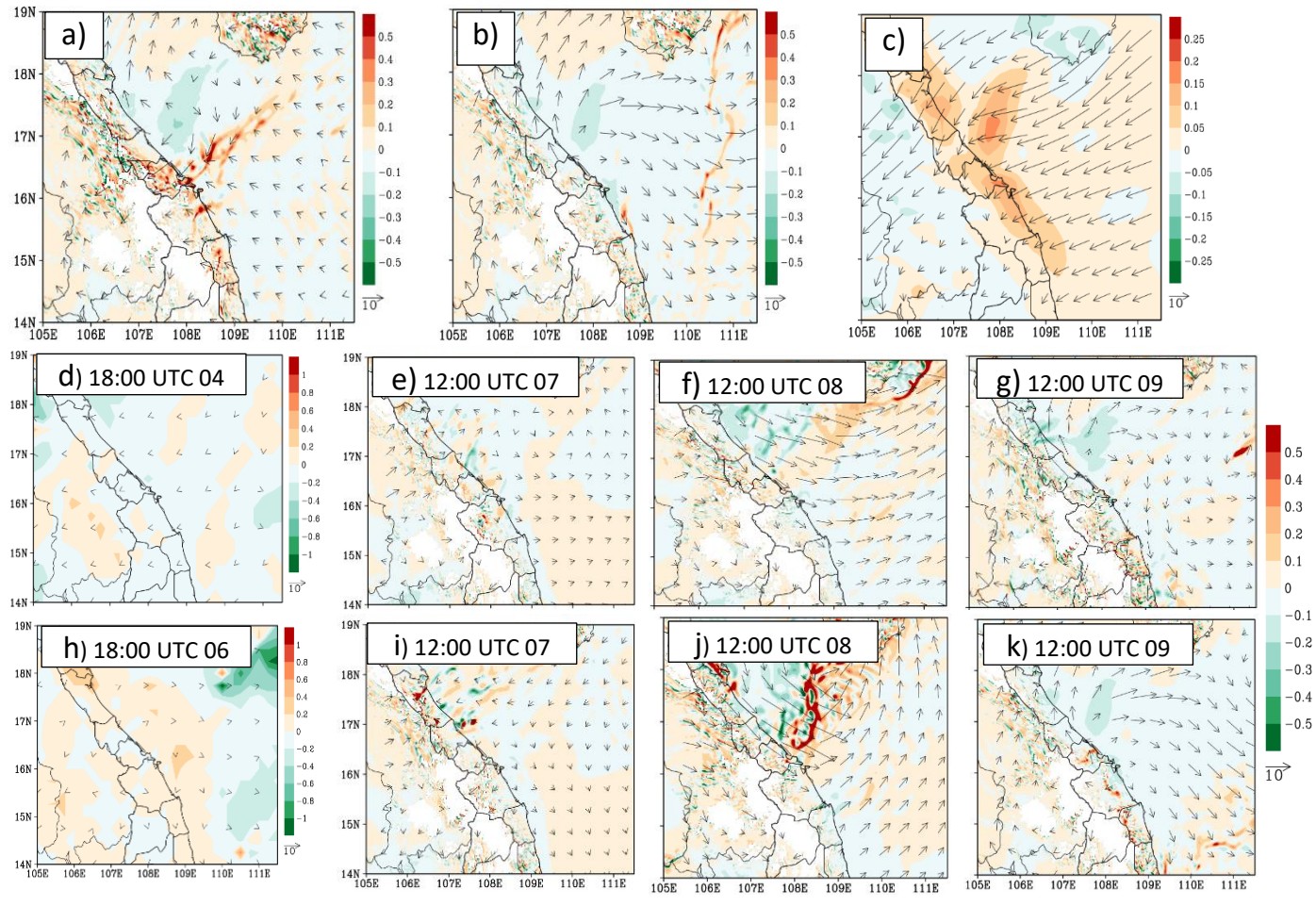


**Figure 10**. The difference in the horizontal wind (ms$^{-1}$, vector, reference length at the low-right corner of the panel), and vertical velocity (ms$^{-1}$, shaded, the reference color scale is on the right of panel) between (a) averaged 5 good members and (b) averaged 5 bad members and the NCEP FNL analysis data at 925 hPa and at 12 UTC 09. (c) The NCEP FNL analysis horizontal wind (ms$^{-1}$, vector, reference length at the low-right corner of the panel) and vertical velocity (ms$^{-1}$, shaded) at 925 mb and at 12 UTC 09. (d) The difference in the horizontal wind (ms$^{-1}$, vector, reference length at the low-right corner of the panel), and relative humidity (%, shaded, the reference color scale is on the right of panel) between the initial data of a good member at a longer lead time (at 1800 UTC 4 Dec) and the NCEP FNL analysis data at 925 hPa. (e), (f), and (g) present the difference in the evolution of weather features with time by this good member. (h) as in (d) but for a bad member (member ran at 1800 UTC 6 Dec). (i), (j) and (k) as in (e), (f), and (g), respectively, but for

mentioned bad member. Compared variables are horizontal wind at 925 hPa (ms$^{-1}$, vector,
reference length at the low-right corner of the panel) and vertical velocity (ms$^{-1}$, shaded,
the reference color scale is on the right of panel). The NCEP FNL analysis horizontal wind
(ms$^{-1}$, vector, reference length at the low-right corner of the panel) and vertical velocity
(ms$^{-1}$, shaded) at 925 mb.
Additionally, the above results also reaffirm that very small differences in the initial data
can lead to a vastly different outcome, especially as the forecast range increases, in extreme
rainfall events (such as the D18 event) that involve highly nonlinear deep convection. As
pointed out in Part 1, the low-level wind convergence led to moisture convergence and
these conditions played a crucial role in the D18 event. The southward movement of the
low-level wind convergence also dictated the movement of the convective rainband during
the event. Therefore, the ESA was applied on relevant variables, including the horizontal
wind and mixing ratio of water vapor. The quantitative results are shown in Figs. 11-13
and presented below.
Figure 11 shows the sensitivity of mean 24-h total rainfall inside the green box in central
Viet Nam ($R$) to zonal ($u$) and meridional ($v$) wind components and water vapor mixing
ratio ($q_v$) at the surface, with the ensemble mean also plotted. It is clear that the sensitivity
of rainfall to these variables is lower at longer forecast ranges and becomes higher as the
lead time shortens. Specifically, from two days before ($t_{-48}$) to the starting time of the
accumulation period ($t_0$), the sensitivity of rainfall to $u$-wind over the SCS and along the
coast of central Viet Nam turned more negative, indicating heavier rainfall associated with
stronger easterly winds ($u < 0$) near the surface, especially in areas immediately upstream
toward $t_0$ (Figs. 11a-d). The rainfall's sensitivity to $v$-wind leading to $t_0$, on the other hand,
exhibited a dipole structure in pattern, with negative values to the north-northwest and
positive values to the south-southeast across central Viet Nam and the upstream ocean
(Figs. 11e-h). This structure indicates a stronger confluence in northeasterly winds over the
region in rainier members, consistent with the results in Part 1. In Figs. 11e-h, the increase
in $v$-wind just south of central Viet Nam is particularly evident, from $-10$ mm per SD (SD
$= 2$ ms$^{-1}$) at $t_{-48}$ to over $+70$ mm per SD (SD $= 2\text{-}4$ ms$^{-1}$) at $t_0$. Thus, the precipitation amount
over central Viet Nam in the D18 event is highly sensitive to the strength and confluence
of northeasterly winds near the surface in short-range forecasts. Similarly, the rainfall was
also highly sensitive to the water vapor amount and its flux convergence (Figs. 11i-l).

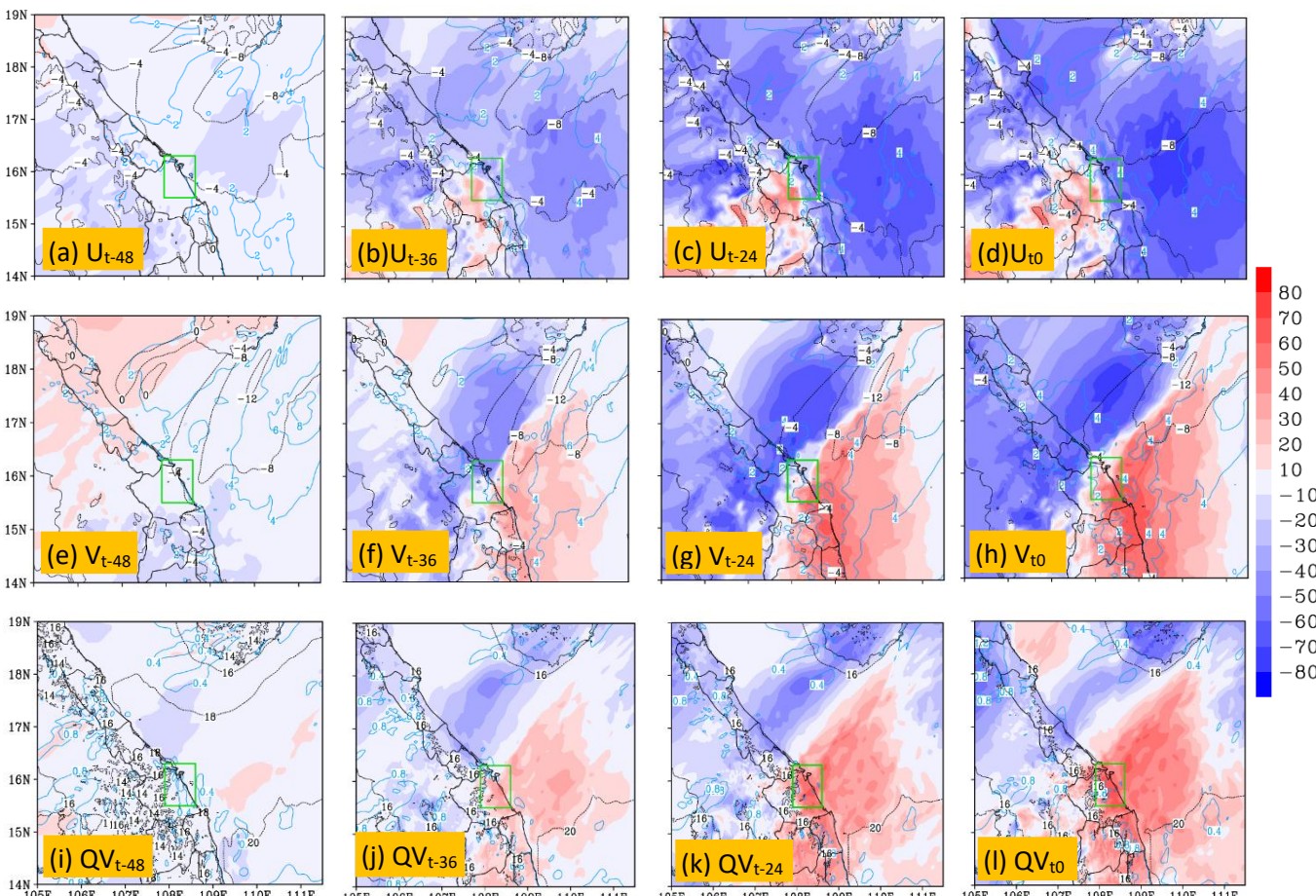

**Figure 11**. The sensitivity (mm, per SD, color, scale on the right) of areal-mean 24h
accumulated rainfall in central Viet Nam starting from $t_0$ (i.e., R, averaging area depicted
in green box) to surface wind components (ms$^{-1}$, shaded) and the ensemble mean (contours,
every 4 ms$^{-1}$) and to surface water vapor mixing ratio (r, g kg$^{-1}$) and its ensemble mean
(contours, every 0.06 g kg$^{-1}$) at different times at 24h intervals from (a) $t_{-48}$ to (f) $t_0$. The
time of $t_0$ is 12:00 UTC 9 December 2018. In which, (a), (b), (c), (d) for the zonal wind
component. (e), (f), (g), (h) for the meridional wind component, and (i), (j), (k), (l) for
surface water vapor mixing ratio. The standard deviation is exhibited by the medium blue
contours.
Slightly higher up at 1476 m (near 850 hPa), where easterly flow prevailed during the D18
event (see Fig. 3b in Part 1), the sensitivity of rainfall to *u* and *v* winds exhibits similar
spatial patterns (Figs. 12a-h) to those at the surface (Figs. 11a-h), with stronger easterly
winds and larger confluence in association with heavier rainfall. Similarly, the rainfall in
central Viet Nam is still highly sensitive to mixing ratio at this level, both locally and over
the surrounding area scale (Figs. 12i-l), again especially at shorter lead times. At the local
scale, this positive correlation presumably is linked to upward transport of moisture, as the
ascending motion in convective clouds could become larger at this level (and also more
vigorous in rainier members).

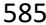

**Figure 12**. The sensitivity (mm, per SD, color, scale on the right) of 24h accumulated rainfall in central Viet Nam starting from $t_0$ (i.e., R, averaging area depicted in green box) to the wind components (ms$^{-1}$, shaded) and the ensemble mean (contours, every 2 ms$^{-1}$) and to water vapor mixing ratio (r, g kg$^{-1}$) and its ensemble mean (contours, every 0.4 g kg$^{-1}$) at attitude of 1476 m and at different times at 24h intervals from (a) $t_{-48}$ to (f) $t_0$. The time of $t_0$ is 12:00 UTC 9 December 2018. In which, (a), (b), (c), (d) for the zonal wind component. (e), (f), (g), (h) for the meridional wind component, and (i), (j), (k), (l) for water vapor mixing ratio. The standard deviation is exhibited by the medium blue contours.

At the upper level of 5424 m (near 500 hPa), it is seen that from $t_{-48}$ to $t_0$, dipole structures developed in the sensitivity patterns of rainfall to both $u$ and $v$ winds (Figs. 13a-h). To $u$ winds, positive sensitivity up to about +70 mm per SD (SD = 2-4 ms$^{-1}$ depending on $t$) existed to the south, with negative values up to −70 mm per SD (SD = 2-4 ms$^{-1}$) to the north of central Viet Nam. Meanwhile, positive sensitivity to $v$-wind appeared to the north and east with negative sensitivity to the south and west of the rainfall area. As the prevailing winds at 500 hPa were southeasterlies over southern Viet Nam and southwesterlies over northern Viet Nam during the D18 event (thus with anticyclonic curvature, see Fig. 3c in Part 1), the above sensitivity patterns, already apparent at $t_{-24}$ (Figs. 13c,g), corresponded to stronger diffluence/divergence and a weaker anticyclone aloft to favor more rainfall. To $q_v$, positive sensitivity signals up to +70 mm per SD (SD = 1.2 g kg$^{-1}$) also appeared over the rainfall area at $t_{-24}$ and $t_0$ (Figs. 13i-l), and the reason is similar to that near 850 hPa in Fig. 12. Overall, the ESA performed in this study indicated clearly that the synoptic pattern that caused the D18 event already developed at times more than 24 h earlier, and this explains why, with a high enough resolution and cloud-resolving capability, the CReSS forecasts could better predict and improve the QPFs inside the short range as shown in Section 3.

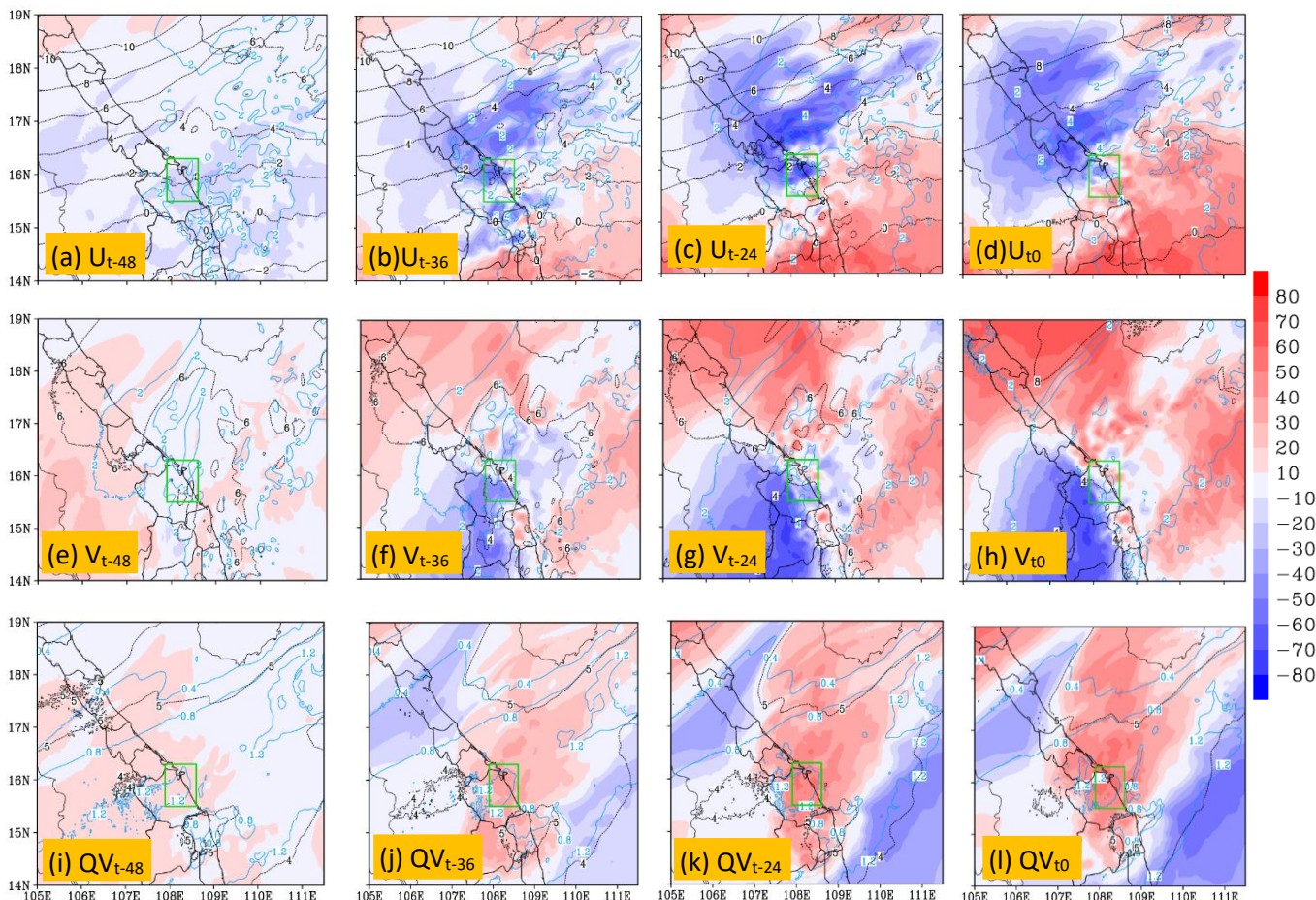

**Figure 13**. The sensitivity (mm, per SD, color, scale on the right) of 24h accumulated rainfall in central Viet Nam starting from $t_0$ (i.e., R, averaging area depicted in green box) to the wind components (ms$^{-1}$, shaded) and the ensemble mean (contours, every 2 ms$^{-1}$) and to water vapor mixing ratio (r, g kg$^{-1}$) and its ensemble mean (contours, every 0.4 g kg$^{-1}$) at attitude of 5424 m and at different times at 24h intervals from (a) $t_{-48}$ to (f) $t_0$. The time of $t_0$ is 12:00 UTC 9 December 2018. In which, (a), (b), (c), (d) for the zonal wind component. (e), (f), (g), (h) for the meridional wind component, and (i), (j), (k), (l) for water vapor mixing ratio. The standard deviation is exhibited by the medium blue contours.

## 4 Conclusion

As high resolution is required in numerical models to predict heavy rainfall more successfully, the present work utilizes a time-lagged high-resolution ensemble forecast

system and evaluates how well the D18 event (during 9-12 December 2018) in central Viet Nam can be predicted in advance before its occurrence. Using the CReSS model with a grid size of 2.5 km ($912 \times 900$ in dimension with 60 vertical levels), ensemble forecasts were produced with a total of 29 time-lagged runs at 6-h intervals, each out to a forecast range of 192 h (eight days). Based on the goals raised from the analysis in Part 1, the key findings of this Part 2 study are summarized as follows:

The first goal of this study is regarding the scientific hypotheses that at a higher resolution, the cloud-resolving time-lagged ensemble can improve the QPFs of the D18 event at the short range, and may also be able to extend the lead time of decent QPFs beyond the short range. Our evaluation results confirm that this strategy using the CReSS model can effectively improve the QPFs of this event at the short range. Furthermore, the results also demonstrate that a decent QPF for 10 December (the rainiest day) can be made at a longer lead time (initialized at 1800 UTC 4 December), when good initial conditions are provided.

About the second goal, our investigation in predictability indicates that the 2.5-km system predicted the rainfall fields on 10 December during the event fairly well, including both the amount and spatial distribution, within the short range at lead times of day 1, 2, and 3. More specifically, the SSS of QPFs at these three ranges are about 0.4, 0.6, and 0.7, respectively, with fairly consistent results among successive runs that indicate a reasonable predictability, despite some spread and disagreement on the precise locations of heavy rainfall. The above good results are due to the model's capability to better predict the conditions in the lower troposphere such as the wind fields.

At lead times longer than three days, however, the predictability of the event is lowered due to a higher level of forecast uncertainty, and the quality of QPFs drops with significant under-prediction. Nevertheless, good QPFs are still possible occasionally. At lead time beyond six days, it is challenging to achieve a good QPF at thresholds greater than 100 mm even with a high-resolution model. This is presumably linked to the rapid evolution of atmospheric conditions during such an extreme event surrounding Viet Nam in a tropical environment. In the present study, a CRM is applied to forecast extreme rainfall in central

Viet Nam for the first time. Although still with certain limitations, our results do indicate
hope to predict such events successfully beforehand, at least within the short range.
Therefore, based on the present work, more studies on the predictability of extreme rainfall
in Viet Nam are recommended in the near future.
Regarding the third and final goal, ESA results show that the rainfall is most sensitive to
the wind conditions in the lower troposphere leading to the event, with more rain associated
with stronger northeasterly to easterly winds and their confluence over central Viet Nam
(and the upstream region). Similarly, the rainfall also shows strong sensitivity to the
moisture amount, not only at the surface but also further aloft at the upper levels. Besides,
ESA also indicates that the synoptic pattern that caused the D18 event already developed
at timing earlier in the past. Furthermore, in the ESA, the finer-scale features (convection)
are also seen to link to synoptic conditions in their background, implying that it is
meaningful to apply ESA to control the perturbations in initial fields.
The key findings in this study underscore that both practical predictability and ESA are
intertwined, influencing the design and evaluation of ensemble forecast systems, and
potentially applicable to other extreme rainfall events in the same season in Vietnam.

*Acknowledgements:* This study was supported by the project "*Research on the application*
*of the Cloud-resolving model integrated with the regional numerical model to a 6-hour*
*accumulated quantitative precipitation forecast with 24-48 hours lead time for Mid-*
*Central Viet Nam*", which is funded by the Ministry of Natural Resources and Environment
(MONRE) under grant no. TNMT.2023.06.07, and also by the National Science and
Technology Council (NSTC) of Taiwan under grants MOST 111-2625-M-003-001, NSTC
112-2625-M-003-001, NSTC 113-2625-M-003-001, and NSTC 113-2111-M-003-001.
*Code and data availability*. The CReSS model used in this study and its user's guide are
available at the model website at http://www.rain.hyarc.nagoya-
u.ac.jp/~tsuboki/cress_html/src_cress/CReSS2223_users_guide_eng.pdf (last access: 6
July 2023; Tsuboki and Sakakibara, 2007). The TIGGE data and its information are
available at https://confluence.ecmwf.int/display/TIGGE/TIGGE+archive. The NCEP
GFS dataset and its description are available at https://rda.ucar.edu/datasets/ds084.1/. The
NCEP FNL operational global gridded analysis data and its information is available at
https://rda.ucar.edu/datasets/d083003/#.
*Author contributions*. **DVN** prepared datasets, executed the model experiments, performed
the analysis, and prepared the first draft of the manuscript. **CCW** also prepared the first
draft and provided the funding, guidance and suggestions during the study, and they
participated in the revision of the manuscript. **KBT** provided the funding and participated
revising of the manuscript. **TVV**, **PTTN**, and **PYC** also participated in the revision of the
manuscript.
*Competing interests*. The authors declare that they have no conflict of interest.

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
