# Peer review of "Investigation of an extreme rainfall event during 8–12 December 2018 over central"

_Natural Hazards and Earth System Sciences, 2023_

## Referee Comment (RC2)

**Review of NHESS-2023-192**

**Title:** Investigation of an extreme rainfall event during 8-12 December 2018 over central Viet Nam – Part 2: An evaluation of predictability using a time-lagged cloud-resolving ensemble system

**Authors:** Chung-Chieh Wang, Duc Van Nquyen, Thang Van Vu, Pham Thi Thanh Nga, Pi-Yu Chuang, and Kien Ba Truong

**Summary:** This manuscript is a follow on to another manuscript that evaluates an extreme rainfall event that occurred in central Vietnam in 2018. The focus of this manuscript is on an evaluation of the predictability of the event. The authors present an analysis of a time lagged ensemble and perform ensemble sensitivity analysis. The authors conclude that predictability increased as lead time decreased for the event. Additionally, they find various atmospheric features are important to the predictability of the storm. While the case is interesting, I find the analysis superficial and incomplete. The overarching results is that an event becomes more predictable when it gets close to occurring, which is well known and does not add to the body of literature on ensemble prediction. For these reasons, I inform a decision of *Reject*. Should the editor or other reviewers come to another conclusion, I am happy to review the manuscript again.

**Recommendation:** Reject

**Substantial Comments (Comments are not listed in order of importance):**
1. I find the evaluation of predictability for this event incomplete. Grouping ensemble members together is a very superficial evaluation of predictability that does not go into the depth needed to examine the actual cause of the lack of predictability. The ensemble sensitivity analysis is a logical next step, but it is simply presented in the manuscript. The results are not interpreted or physically linked back to the event. They are simply presented.
    a. The result of predictability increases as lead time decreases is not a new result to the body of literature on meteorological prediction.
    b. An analysis of the differences, physically, between each ensemble run that might be the cause of the lack of predictability should be undertaken. This should be more than just low-level RH and surface winds, as it is well known that large scale features are important to controlling these factors (see the results from Part 1). This will then lend context to the ESA and identify how these sensitivities feedback into the prediction.
2. No hypotheses are presented in this work. This leads to the manuscript being unorganized, and the results unclear in the context of the broader literature. Having model simulations are not alone publishable. It is thus important to outline scientific based hypotheses in which the experiments in the manuscript are designed to evaluate, which will then make it clearer how the work adds to the body of literature.
3. Ensemble spread is not purely error or a representation of accuracy. The goal of a well calibrated ensemble is to represent the forecast probability density function. Thus, if there is high uncertainty, we want the ensemble to have a large amount of spread. If there is

small uncertainty in the system, we want the ensemble to have little spread. The usage of spread as an error metric needs to be done within this context.

   a. It is also not clear to me where the spread analysis is undertaken within the paper.
   b. Some discussion and framing of the work here from a context of intrinsic versus practical predictability is needed. Additionally, the scale dependence of predictability. I suggest Melhauser and Zhang (2012), Nielsen and Schumacher (2016), Weyn and Durran (2018), and citations within as starting points. There is also some useful suggestions from an ensemble analysis within these papers.

4. The results presented in this paper would have made an interesting section in Part 1 paper but because of these issues outline above it is not in the current state publishable on their own.

**Additional Comments (Comments are not listed in order of importance):**

1. Lines 21: typo "predicts"
2. Lines 53-55: The phrasing of this sentence is awkward. I recommend removing "until now" and adding something like "to improve predictability" to the end of the sentence.
3. Lines 85-87: Citations are needed to support this statement.
4. Lines 182-184: What version of the GFS?
5. Section 2.1.3: Is there any citation or information about the observational error associated with the gauges in this network?
6. Section 2.1.4: A citation needs to be added for IMERG. Additionally, please let us know what version you used.
7. Line 247: Table 2 does not appear anywhere in the manuscript.
8. Lines 485-488, Lines 508-509: It is also possible the convective inflow to the storms is not at the surface but is elevated. Again, a much more detailed examination into the variables that control the ingredients for extreme rainfall is needed in the ensemble runs and ESA.
9. Section 3.2: What does "per SD" mean?

References:

https://journals.ametsoc.org/view/journals/atsc/69/11/jas-d-11-0315.1.xml?tab_body=fulltext-display

https://journals.ametsoc.org/view/journals/mwre/144/10/mwr-d-16-0083.1.xml?tab_body=fulltext-display

https://rmets.onlinelibrary.wiley.com/doi/full/10.1002/qj.3367

---

## Author Response (AR1)

NHESS-2023-192

**Authors' Responses to Reviewer 1 (RC1, anonymous)**

Date: 2 September 2024

Title: Investigation of an extreme rainfall event during 8–12 December 2018 over central Viet Nam – Part 2: An evaluation of predictability using a time-lagged cloud-resolving ensemble system

Authors: C. C. Wang et al.

Firstly, **we thank the reviewer for spending valuable time reviewing the paper and giving us constructive comments that helps to improve the clarity of the paper.**

**COMMENTS**

In general, I think this is a good follow-up study to evaluate the usefulness of CReSS ensembles in forecasting an extreme rainfall event over Viet Nam. However, I believe significant editorial improvements (texts and figures) are necessary to enhance the manuscript's readability. For example, varies data are described in the data section (Section 2.1), but many of them are not used in the subsequent sections (e.g. TIGGE, WRF). Some of the colour bars and their labels are quite small. Overall, I think the current manuscript needs major revision.

Specific comments:

Abstract: The authors should state the full form first before using abbreviations e.g. CReSS, QPF, SSS, and ESA.

**Reply:** Thank you for your comment. We have made corrections and give the full form in the first use of these abbreviations. The corrections are shown on page 1, lines: 17, 20 and 21, in the marked-up manuscript version.

Line 36: Perhaps it would add some clarity to "The observational data …" by adding GPM in the text.

**Reply:** Thanks for your suggestion. Here, we are referring to the rain-gauge observations, so we mention this explicitly to add some clarity, along the lines as

suggested. The GMP data are also used in this study, so they are mentioned later in the text.

Lines 40-42: Missing reference(s)?

**Reply:** We agree with you, we have added the reference as suggested. The reference was added on page 2, line 44, in the marked-up manuscript version.

Lines 97-98: Missing reference(s)?

**Reply:** Thank you for your concern. The references were cited on line 101 in the previous draft, and we now move some of them here for better fluency and clarity, along the lines as suggested. The references were added on page 5, line 103, in the marked-up manuscript version.

Lines 113-119: I am not sure whether these details are relevant to the current study.

**Reply:** Thank you for your comments. To avoid too much detail, we have deleted the specific numbers in the favorable factors in the revision, as suggested. The changes are shown on page 6, lines 119-124, in the marked-up manuscript version.

Lines 123-139: Given the nature of this study is to demonstrate the CReSS ensemble can produce good forecast of extreme precipitation events. Perhaps it is necessary to have a brief description of other models, which could perform simulation at a cloud-resolving resolution, e.g. MM5 by Son and Tan (2009), WRF by Toan et al. (2018) and Nhu et al. (2017). Information such as resolution and relevant parameterisation schemes (if any) would be relevant in this case. If the resolution and relevant parametrisation schemes of those models would be the same as the current study, the authors might want to emphasis this point.

**Reply:** Thanks for your suggestion. We will make changes and add the brief description in the revision, as suggested. The changes are shown on pages 6 -7, lines 130-132, lines 136-139, and lines 146-147, in the marked-up manuscript version.

Lines 139-147: The authors might want to highlight the fact that these global NWP models do not have "cloud-resolving resolution".

**Reply:** Yes, you are right and we will make the highlight as suggested. Besides, we would also like to point out that the current global weather prediction models cannot capture the precipitation at the correct quantity in Vietnam. The added

information is shown on page 7, lines 158-160, in the marked-up manuscript version.

Line 173: Not sure why UKMO is mentioned but their forecast outputs were not used. Also "ECMWF of the European Union" is not accurate as the UK is a member state of ECMWF but not in the European Union.

**Reply:** Thank you for pointing this out. We have removed that information. We used the UKMO data in the first version of the manuscript. However, we had removed it from the submitted version of manuscript. But we were missed to remove the relevant information at line 173. The changes are shown on page 10, lines 218-219, in the marked-up manuscript version.

Lines 175-176: The authors should state the usage of this dataset at the beginning of the paragraph first rather than in the middle of the paragraph.

**Reply:** Thanks for your suggestion. In the revision, we have mentioned that this dataset is used in the study in the first sentence of the paragraph, as suggested. The changes are shown on page 9, lines 212-213, in the marked-up manuscript version.

Section 2.1.2: It might be worthwhile to highlight the fact that NCEP GFS is a deterministic forecast.

**Reply:** Thank you for your comments. As suggested, this clarification has been made in the revision and now the sentence reads as "… from the analyses and deterministic forecast runs…" the change is shown on page 12. Lines 272-273, in the marked-up manuscript version.

Section 2.1.3: The authors should highlight the fact that these are the in-situ observation data as GPM data is also a type of "observation" data.

**Reply:** Thank you for your suggestion. We will make change in the revision, as suggested. The changes are shown on page 8, lines 181-182, in the marked-up manuscript version.

Section 2.1: The authors might want to restructure this section so that the data, which serves as similar purpose would be grouped together. A possible structure could be: "Section 2.1.1" In-situ observation data and "Section 2.1.2" GPM are used to model validation (ground truth); "Section 2.1.3" TIGGE and "Section

2.1.4" WRF are used to demonstrate the added values of CReSS ensemble; "Section 2.1.5" NCEP GFS is used to drive CReSS. This structure would fit nicely into "Section 2.2 Model description …"

**Reply:** Thank you for your suggestions. We will make change in the revision, as suggested. The changes are shown on pages 8 -12, in the marked-up manuscript version.

Figure 1, Section 2.1: ERA5 was used in Figure 1 but was not mentioned in Section 2.1.

**Reply:** Thank you for your comments. After carefully checking, we found that Fig. 1f where ERA5 data are used, has been adapted directly from Fig. 14c of Wang and Nguyen 2023, and we also stated this information in the caption of Figure 1. Therefore, we did not mention it in section 2.1. if our explanation is not legal, we are happy to make the change, in the marked-up manuscript version.

Lines 228-230: "The first members ran at 12:00 UTC on 3 December 2018, and the last member ran at 12:00 UTC on 10 December 2018…" à "The first member was initialised at 12:00 UTC on 3 December 2018, and the last member was initialised at 12:00 UTC on 10 December 2018. A new member was initialised every 6-hr within the period 1200 UTC 3 Dec 2018-1200 UTC 10 Dec 2018."

**Reply:** Thank you for your suggestions. We will make change in the revision, as suggested. The changes are shown on page 12, lines 281-285, in the marked-up manuscript version.

Section 2.1.2, Lines 232-234, and Table 1: I am a bit confused. Did the authors use NCEP FNL (Lines 232-234, Table 1; also, from the first part of this two-part study [Wang and Nguyen 2022]) or NCEP GFS (as stated in Section 2.1.2) to drive CReSS?

**Reply:** Thank you for your comments. We corrected the information. It is NCEP GFS data as stated in Section 2.1.2. The correction is shown on page 12, lines 288-289, in the marked-up manuscript version.

Line 278: "If small spread …" à "For example, small spread indicates…"

**Reply:** Corrected. The correction is shown on page 16, line 336, in the marked-up manuscript version.

Figure 3, Lines 312-313, Figure 5: I believe the OBS, which the authors are referring to, is not from GPM nor in-situ observations but from ERA5 as it has more similarity to Figure 1f than Figure 1e. The authors also mentioned stations but stations are not indicated in Figure 3. Perhaps, the authors meant to show another figure?

**Reply:** We apologize for the confusion. The data shown in Fig. 3 OBS is a combination of 24-h observed rainfall from the rain-gauge network and the surface wind analysis from ERA5, and the same is shown in Fig. 1f. We have clarified the information in the caption of both Figs. 1 and 3.

Figure 3: Perhaps I missed it, but it is not clear to me how good members (green) and bad members (red) are determined. For example, why does "00:00 UTC 9" is a good member whereas "18:00 UTC 8", "06:00 UTC 9", and "12:00 UTC 9" are not classified as good members but the spatial structures of these members are very similar to "00:00 UTC 9".

**Reply:** We apologize for the confusion. In this study, good and bad members are determined by both visual comparison and the SSS skill score. Specifically, we compared the model results with observation, and good members were picked if the quantitative rainfall and the spatial distribution of the rain band were closer to observation. For the skill score, we applied the similarity skill score to all members and picked good or bad members based on their score. Higher score means better members and lower score means worse member.

[Figure]

**Figure 1.** Similarity skill score of 25 time-lagged members for the 24-h accumulated rainfall of the Dec 10, 2018

Lines 318, 324: "… executed …" à "… initialised …"

**Reply:** Corrected. The correction is shown on page 17, lines 377-378, and 382, in the marked-up manuscript version.

Line 341: Is it GFS or FNL?

**Reply:** Here, we are referring to the GFS forecast data, so it is the GFS instead of the FNL data, which is the analysis. Specifically, it is GFS version ds084.1. We hope that this is clear.

Lines 353-367; Figure 5: (1) Perhaps it might be clearer if the authors would rename the subgroups by using the period of forecast initialisations, e.g. "First 4 members" à "12 UTC 3- 06 UTC 4 Dec". (2) Similar grouping could be done by defining period using a "moving window period", e.g. "Subgroup 1: 12 UTC 3- 06 UTC 4 Dec", "Subgroup 2: 18 UTC 3- 12 UTC 4 Dec" etc.. The "moving window" approach could help the authors to better locate the optimal initialisation periods. (3) If the "moving window" approach is used, perhaps some kind of measure of the deviation of the 24h rainfall field between OBS and subgroups would be useful in quantifying the optimal initialisation periods.

**Reply:** Thank you for your comments and suggestions. We will add more information to our analysis to make it clearer. The added information is shown on page 20, lines 417, 421, and 424, in the marked-up manuscript version.

Line 365: The rainfall of second 4 members is the highest "in" these… (A word is missing)

**Reply:** Thank you for your careful review. We have corrected it, as suggested. The correction is shown on page 20, line 424, in the marked-up manuscript version.

Line 366: … due to "a single good forecast initialised at 1800 UTC on 4 Dec".

**Reply:** Thank you for your careful review. We have corrected it, as suggested. The correction is shown on page 20, line 425, in the marked-up manuscript version.

Lines 384-388: The definition of subgroups seems to be arbitrary. It is not clear to me why these subgroups are chosen for in-depth analysis. Furthermore, would this not be expected that the forecasts initialised closer to the target period would have a better forecast skill as certain features that are highly related to the extreme precipitation event are included in the initial conditions? In this sense, an interesting question would be: Why does the forecast initialised on 1800 UTC 4 Dec can (partially) capture the spatial extent of the extreme precipitation event of interest and such information was lost for the next 72 hours?

**Reply:** Our purpose in this paragraph is to examine how forecast quality evolved as the lead time shortened. Therefore, along with the evaluation on time-lagged results using batches of fixed number of successive runs (every 4 members) as presented in our analysis, this study also grouped the members using different ensemble sizes based on their behavior in order to better assess the temporal evolution of forecast uncertainty and event predictability as the lead time shortened

Figures 10, 11, 12: The authors should add some labels (u, v, qv) on the plot. Also missing labels on Figure 10a and Figure 10c

**Reply:** Thank you for your comments and suggestions. We will add this information to our figures to make it clearer. The added information is shown in Figures 11-13, in the marked-up manuscript version.

Figure 10: Is it relating to surface wind, which typically is defined as 10-m wind, or 100-m wind (as stated in Lines 473-474).

**Reply:** Thank you for your comments. After carefully checking, we found that the mentioned data is exactly at the surface level. We corrected the information and this correction is shown on page 31, line 599, in the marked-up manuscript version.

Section 3.2: The authors should indicate which graph/panels the readers should be looking for. E.g. Lines 476-479: It should be referring to Figure 10a-c. etc.

**Reply:** Thank you for your comments and suggestions. Here, we are referring to wind components in Figures. 10a-f. We have added more information to our analysis to make it clearer. The added information is shown on pages 31-32, lines 605, 609, and 615, in the marked-up manuscript version.

Lines 502-509: I am confused about the Figures and the texts. What levels are actually showing and the authors are referring to. I guess the authors are comparing Figures 11 and 10? If this is the case, Line 504 should include something like "… 100 meters (Figure 10).

**Reply:** Thank you for your comments and suggestions. We also apologize for the confusion. Here, we are referring to wind components and water vapor mixing ratio at the surface. Therefore, the Figure referred to is Figure 11, as you guessed. we have added more information to our analysis to make it clearer. The added information is shown on page 33, lines 641, in the marked-up manuscript version.

**Authors' Responses to Reviewer 2 (RC2, anonymous)**

**COMMENTS**

Substantial Comments (Comments are not listed in order of importance):

**Comment 01.** I find the evaluation of predictability for this event incomplete. Grouping ensemble members together is a very superficial evaluation of predictability that does not go into the depth needed to examine the actual cause of the lack of predictability. The ensemble sensitivity analysis is a logical next step, but it is simply presented in the manuscript. The results are not interpreted or physically linked back to the event. They are simply presented.

**Reply:** Thank you for your comments. We are reinstructing our paper to make our current study more logical and clearer. Specifically, we moved Fig. 4 to section 3.2, adding more new results, as well as put more emphasis on the analysis to (1) better clarify the predictability of D18 event, and to (2) present the new findings in our study more effectively and more clearly. The change is shown on pages 19-20, lines 393-409, and in section 3.2, in the marked-up manuscript version.

a. The result of predictability increases as lead time decreases is not a new result to the body of literature on meteorological prediction.

**Reply:** You are right. However, our results once again reconfirm that predictability decreases as lead time increases in general, especially for a region with complex terrain and weather systems like Vietnam. Furthermore, in our study, it is shown that the lead time is not the only factor to influence the predictability (as one may suspect). Specifically, there is a good forecast at a longer lead time, made by the member ran at 1800 UTC 4 Dec. This indicates the potential for heavy rainfall to occur with more time for preparation. We will put more emphasis on this member in the revision, and conclude that it is still possible to have good forecasts at a lead time up to 5 days. The deeper analysis is shown in section 3.2. pages 29-31, in the marked-up manuscript version.

b. An analysis of the differences, physically, between each ensemble run that might be the cause of the lack of predictability should be undertaken. This should be more than just low-level RH and surface winds, as it is well known that large scale features are important to controlling these factors (see the results from Part 1). This will then lend context to the ESA and identify how these sensitivities feedback into the prediction.

**Reply:** Thank you for comments. Based on the results from Part 1, the low-level wind convergence led to moisture convergence and these conditions resulted in the D18 event. Furthermore, the southward movement of the low-level wind convergence also dictated the movement of heavy rainband during the event. Therefore, we would like to put more focus on these two aspects in the revision.

To clarify your concern, we are plotting more results from NCEP FNL analysis data. We would like to compare the evolution of synoptic-scale patterns (features) in the sole good member at a longer lead time (at 1800 UTC 4 Dec) and bad member with that in NCEP FNL analyses to better link the performance of CReSS runs back to the physics, and also better point out a new interesting result in this study. that is still possible to have good forecasts at lead time up to 5 days. The new result is Figure 10, in the marked-up manuscript version.

Besides, we also plot the sensitivity at t-36 to better identify the timing with higher predictability. Because we found a new interesting result (new finding) in section 3.2 that the synoptic pattern already developed into (or toward) what would cause the rainfall later at timing more than 24h earlier. We believe that these could explain the predictability of D18 event. The plots for the sensitivity at t-36 are added in Figures 11-13, in the marked-up manuscript version.

**Comment 02**. No hypotheses are presented in this work. This leads to the manuscript being unorganized, and the results unclear in the context of the broader literature. Having model simulations are not alone publishable. It is thus important to outline scientific based hypotheses in which the experiments in the manuscript are designed to evaluate, which will then make it clearer how the work adds to the body of literature.

**Reply:** As we replied to your comment 01 that we are reorganizing the manuscript to better explain the predictability of D18 event as well as the performance of the CReSS model. We also add more results in the revision, as replied to comment 01, to have deeper look at interesting results (new findings), including (1) it is still possible to have good forecasts at longer lead time (up to 5 days) for record-break events like D18 event. (2) the development of synoptic patterns into (or toward) what could lead to an extreme rainfall later at timing earlier in the past. By these changes, we believe that our present study will have valuable contributions to the body of literature. The change is observed in the marked-up manuscript version.

**Comment 03**. Ensemble spread is not purely error or a representation of accuracy. The goal of a well calibrated ensemble is to represent the forecast probability density

function. Thus, if there is high uncertainty, we want the ensemble to have a large amount of spread. If there is small uncertainty in the system, we want the ensemble to have little spread. The usage of spread as an error metric needs to be done within this context.

a. It is also not clear to me where the spread analysis is undertaken within the paper.

**Reply:** We apologize for the confusion. In the time-lagged approach, only one run is executed at each initial time, so for each time it is not possible to derive the probability information. However, when successive runs are grouped together, one can see that the spread (e.g., standard deviation) still evolves with time just like a multi-member ensemble, as shown in Fig. 8. As the lead time shortened, the high SD region from the last several members became more focused along the coast, indicating high rainfall amounts (also in Fig. 9) there but with uncertainty in exact locations.

b. Some discussion and framing of the work here from a context of intrinsic versus practical predictability is needed. Additionally, the scale dependence of predictability. I suggest Melhauser and Zhang (2012), Nielsen and Schumacher (2016), Weyn and Durran (2018), and citations within as starting points. There is also some useful suggestions from an ensemble analysis within these papers.

**Reply:** Thank you for your comments and your information. We will add more discussion (as mentioned above) and cite these references in the revision.

**Comment 04**. The results presented in this paper would have made an interesting section in Part 1 paper but because of these issues outline above it is not in the current state publishable on their own.

**Reply:** Thank you for the comment. As it is, the Part I paper has already been published, and we certainly would like to published the high-resolution time-lagged QPF result on the D18 event as Part II, if at all possible. We will devote efforts to clarify the predictability, factors affecting the predictability, and the usefulness of the time-lagged ensemble in this event. We hope that our reply (including those stated above) and future revision will shed light on the above issues, be satisfactory, and make contributions to the existing literature on heavy-rainfall QPFs.

Additional Comments (Comments are not listed in order of importance):

1. Lines 21: typo "predicts"

**Reply:** Thank you for your comment. We corrected it. The correction is shown on page 1, line 22, in the marked-up manuscript version.

2. Lines 53-55: The phrasing of this sentence is awkward. I recommend removing "until now" and adding something like "to improve predictability" to the end of the sentence.

**Reply:** Thank you for your suggestion. We updated it, as suggested. The change is shown on page 3, line 57, in the marked-up manuscript version.

3. Lines 85-87: Citations are needed to support this statement.

**Reply:** Thank you for your suggestion. We added the information, as suggested. The added information is shown on page 5, line 91, in the marked-up manuscript version.

4. Lines 182-184: What version of the GFS?

**Reply:** we used the GFS version ds084.1. We also added the information in the revision for clarification. The added information is shown on page 12, line 272, in the marked-up manuscript version.

5. Section 2.1.3: Is there any citation or information about the observational error associated with the gauges in this network?

**Reply:** Thank you for your question. The automated rain-gauge network in Vietnam is operated, maintained, and managed by Vietnam National Centre for Hydro-Meteorological Network (NCN). Therefore, the network meets the World Meteorological Organization standard. The observed dataset in this study is provided by NCN through the Mid-Central Regional Hydro Meteorological Center.

6. Section 2.1.4: A citation needs to be added for IMERG. Additionally, please let us know what version you used.

**Reply:** Thank you for your comment. We used the IMERG Final Run V07 data production. We also clarified this and added a citation, as suggested. The change is shown on pages 8-9, in the marked-up manuscript version.

7. Line 247: Table 2 does not appear anywhere in the manuscript.

**Reply:** Thank you for your comments. We had removed that information. We created the table 2 in the first version of the manuscript. However, we had removed it from the submitted version of manuscript. But we were missed to remove the relevant information at line 247. The change is shown on page 14, lines 304-306, in the marked-up manuscript version.

8. Lines 485-488, Lines 508-509: It is also possible the convective inflow to the storms is not at the surface but is elevated. Again, a much more detailed examination into the variables that control the ingredients for extreme rainfall is needed in the ensemble runs and ESA.

**Reply:** You are right. It is necessary to add more detailed examinations. Therefore, we will put more emphasis on the analysis along with adding new results to figure out the quantitative contribution of selected variables to the D18 event. We will let you see it in the revision. The change is shown in section 3.2, in the marked-up manuscript version.

9. Section 3.2: What does "per SD" mean?

**Reply:** We apologize for the confusion. It means per standard deviation. We will add more information along with quantitative values in the revision for clarification. The change is shown on pages 32-35, lines 611, 671-672, 679, in the marked-up manuscript version.

---

## Referee Report (RR1)

**Review of NHESS-2023-192**

**Title:** Investigation of an extreme rainfall event during 8-12 December 2018 over central Viet Nam – Part 2: An evaluation of predictability using a time-lagged cloud-resolving ensemble system

**Authors:** Chung-Chieh Wang, Duc Van Nquyen, Thang Van Vu, Pham Thi Thanh Nga, Pi-Yu Chuang, and Kien Ba Truong

**Summary:** After the first round of revisions, the authors have done a deeper analysis of the results. While I do find the manuscript more cohesive, some of my initial concerns from the first review are still present. These include the lack of presented hypotheses and what this paper adds to the body of literature, chiefly. The additional analysis allows me to see a path to deal with these concerns, compared to the last version of the manuscript. For these reasons, I inform a decision of *Major Revisions*.

**Recommendation:** Major Revisions

**Major Comments (Comments are not listed in order of importance):**

1. No hypotheses are presented in this work, this comment remains the same from the previous review. Please include specific hypotheses. This will also help guide the reader into making clear what the manuscript adds to the body of literature. As I have said before, having model simulations are not alone publishable unless used to evaluate a scientific question/hypothesis. It is thus important to outline scientific based hypotheses in which the experiments in the manuscript are designed to evaluate, which will then make it clearer how the work adds to the body of literature. Please state them.

2. From the last round of revision: Some discussion and framing of the work here from a context of intrinsic versus practical predictability is needed. Additionally, the scale dependence of predictability. I suggest Melhauser and Zhang (2012), Nielsen and Schumacher (2016), Weyn and Durran (2018), and citations within as starting points. There is also some useful suggestions from an ensemble analysis within these papers. The authors mention in the review responses that there was discussion on intrinsic vs. practical predictability added as well as these references. I could not find where in the manuscript this was done.

3. How was the WRF data mentioned in the methods section used? I did not see this in the new version of the manuscript. It is possible I missed it. If it is not used, please remove from the methods.

4. As I have mentioned above and in the last review, there needs to be some aspect of the conclusions/experiments that add to the body of literature ensemble prediction of extreme precipitation. This is currently missing in my opinion, as we generally know predictability increases as lead-time decreases. The authors claim that this is a novel result for this geographic region in their review responses. I can be convinced of this, if this is clearly demonstrated with correctly posed hypotheses and results discussion (see point 1). One potential avenue could be comparing the spread in the event given by a time lagged ensemble compared to a traditional ensemble initialized at one specific time

leading up to the event. This might speak to more general predictability in the region. The above citations can potentially aide in adding this piece to the paper.

**Additional Comments (Comments are not listed in order of importance):**
1. Throughout: Please make use of more concise paragraphs to organize the manuscript. Some paragraphs in the current version span the entirety of a page or more. Breaking these up into smaller paragraphs at logical points will help with the flow and readability of the manuscript.
2. Figures 10-13: These are quite hard to read. I cannot make out the light blue contours that are overlaid on these plots. I can barely make out the black contours. Please evaluate the color choices on these. Additionally, the panels are quite small. It might make sense to break these into multiple figures.
3. Lines 336-338: This is true for a well calibrated ensemble, only. I would mention this.
4. Table 1 and lines 266: How many moments is the bulk microphysics scheme?
5. Line 55: Remove "until now," as it contradicts the rest of the sentence.

---

## Referee Report (RR2)

**Review of NHESS-2023-192**

**Title:** Investigation of an extreme rainfall event during 8-12 December 2018 over central Viet Nam – Part 2: An evaluation of predictability using a time-lagged cloud-resolving ensemble system

**Authors:** Chung-Chieh Wang, Duc Van Nquyen, Thang Van Vu, Pham Thi Thanh Nga, Pi-Yu Chuang, and Kien Ba Truong

**Summary:** After the second round of revisions, the authors have generally satisfied all of my concerns. However, there are still a few more minor comments that I have. For these reasons, I inform a decision of *Minor Revisions*.

**Recommendation:** Minor Revisions

**Minor Comments (Comments are not listed in order of importance):**
1. Line 51: Should this be figure 1, as opposed to figure 4?
2. Lines 98-99: Remove "has become indispensable for its ability to" and change "simulate" to simulates.
3. Lines 106-111: Yes, generally over long periods the ensemble mean is more accurate. However, it should be noted that it will smooth out the precipitation field and likely not capture the extreme magnitude of events.
4. Lines 142-155: There is a scale dependence to the practical and intrinsic predictability as well. Likely worth noting in this section.
5. Lines 156-189: Need to break this into smaller paragraphs. I suggest one at the end of line 172 and 179.
6. Line 256: I would remove the word "retrieval" when talking about the TIGGE data. This leads to some confusion in my opinion, as "retrieval" has connotations of a mathematical processing; not just getting the data.
7. Section 2.3.1: If I am understanding the SSS correctly, it does not take into account neighborhoods when verifying the precipitation? This would be something to specify, if so. Further, using a verification method that allows for a precipitation object to be slightly off location but still have a similar shape would be a useful comparison. Fraction skill score could be an option.
8. Lines 392: "This is true for a well calibrated ensemble, only." Does not seem to go here. I know this was a comment I had from the last round of revisions…but I am not sure why it got placed here.

---

## Author Response (AR2)

NHESS-2023-192

**Authors' Responses to Reviewer 1 (RC1, anonymous)**

Date: 7 January 2025

Title: Investigation of an extreme rainfall event during 8–12 December 2018 over central Viet Nam – Part 2: An evaluation of predictability using a time-lagged cloud-resolving ensemble system

Authors: C. C. Wang et al.

Firstly, **we thank the reviewers for spending valuable time reviewing the paper again and giving us constructive comments that helps to improve the clarity of the paper.**

These are our responses to reviewers:

**Authors' Responses to Reviewer 1 (RC1, anonymous)**

**COMMENTS**

There are quite a bit of formatting issues and typos. I recommend technical corrections.

Here are some examples.

Line 17: Could-resolving --> Cloud resolving

**Reply:** Thank you for your correction. We corrected it. The correction is shown on page 1, line 19, in the marked-up manuscript version.

Line 353: Formatting error.

**Reply:** Thank you for your comments. You are right! We corrected it. The correction is shown on page 20, line 409, in the marked-up manuscript version.

Line 512: Not sure what is happening here. I think the authors might have accidentally deleted some text.

**Reply:** Thank you for your comments. You are right! We got mistake while trying to deleted the text here. We corrected it. The correction is shown on page 32, line 600, in the marked-up manuscript version.

**Authors' Responses to Reviewer 2 (RC2, anonymous)**

Major Comments (Comments are not listed in order of importance):

1. No hypotheses are presented in this work, this comment remains the same from the previous review. Please include specific hypotheses. This will also help guide the reader into making clear what the manuscript adds to the body of literature. As I have said before, having model simulations are not alone publishable unless used to evaluate a scientific question/hypothesis. It is thus important to outline scientific based hypotheses in which the experiments in the manuscript are designed to evaluate, which will then make it clearer how the work adds to the body of literature. Please state them.

**Reply:** Thank you so much for your valuable comments and for reminding us stating our specific hypotheses. We are sorry for forgetting to state it in previous revision. We have added it in this version, on page 10, lines 203-214, in the marked-up manuscript version.

2. From the last round of revision: Some discussion and framing of the work here from a context of intrinsic versus practical predictability is needed. Additionally, the scale dependence of predictability. I suggest Melhauser and Zhang (2012), Nielsen and Schumacher (2016), Weyn and Durran (2018), and citations within as starting points. There is also some useful suggestions from an ensemble analysis within these papers. The authors mention in the review responses that there was discussion on intrinsic vs. practical predictability added as well as these references. I could not find where in the manuscript this was done.

**Reply:** Thank you for your comments. We added more the discussion using your suggested materials to clarify the scientific rationale of our study. Besides, we also cited one more reference (Ying and Zhang 2017) to support our discussion. The added information is shown on pages 7-8, lines 142-154, in the marked-up manuscript version.

3. How was the WRF data mentioned in the methods section used? I did not see this in the new version of the manuscript. It is possible I missed it. If it is not used, please remove from the methods.

**Reply:** We sincerely apologize for your confusion. Initially, we included this information to provide additional details about the mesoscale model (WRF model) whose forecasts were used to discuss the predictability of the D18 event in Fig. 1. However, we have removed this information based on your comments. The change is shown on page 12, lines 268-274, in the marked-up manuscript version.

4. As I have mentioned above and in the last review, there needs to be some aspect of the conclusions/experiments that add to the body of literature ensemble prediction of extreme precipitation. This is currently missing in my opinion, as we generally know predictability increases as lead-time decreases. The authors claim that this is a novel result for this geographic region in their review responses. I can be convinced of this, if this is clearly demonstrated with correctly posed hypotheses and results discussion (see point 1). One potential avenue could be comparing the spread in the event given by a time lagged ensemble compared to a traditional ensemble initialized at one specific time leading up to the event. This might speak to more general predictability in the region. The above citations can potentially aide in adding this piece to the paper.

**Reply:** Thank you very much for your comments. we have adjusted the conclusions part to provide clear answers regarding our scientific hypotheses and the goals of the study based on your comments/suggestions. The change is shown on pages 39-40, lines 712-721, and 740-754 in the marked-up manuscript version.

Additional Comments (Comments are not listed in order of importance):

1. Throughout: Please make use of more concise paragraphs to organize the manuscript. Some paragraphs in the current version span the entirety of a page or more. Breaking these up into smaller paragraphs at logical points will help with the flow and readability of the manuscript.

**Reply:** Thank you for your comments. Based on your comments and suggestions, we have made substantial revisions to the manuscript, including breaking up some paragraphs into smaller sections to improve readability, as you recommended. We believe the manuscript is now slightly shorter and more concise. The change is shown on all pages, in the marked-up manuscript version.

2. Figures 10-13: These are quite hard to read. I cannot make out the light blue contours that are overlaid on these plots. I can barely make out the black contours. Please evaluate the color choices on these. Additionally, the panels are quite small. It might make sense to break these into multiple figures.

**Reply:** Thank you for your comments/suggestions. Unfortunately, we are currently unable to replot the figures due to limitations in accessing the supercomputer. Additionally, we believe that breaking the current panels into multiple figures might make it challenging for readers to follow the evolution of variables over time. To address this, we have removed the latitude and longitude labels from all panels displaying the same region. This adjustment saves space between the panels and allows us to enlarge them slightly. The changes are shown in mentioned figures in the marked-up manuscript version.

3. Lines 336-338: This is true for a well calibrated ensemble, only. I would mention this.

**Reply:** Thank you for your suggestion. We have added this information into the manuscript. The added information is shown on page 19, line 392 in the marked-up manuscript version.

4. Table 1 and lines 266: How many moments is the bulk microphysics scheme?

**Reply:** We are sorry for your confusion. We used a double-moment Bulk cold-rain scheme. We added more information to Table 1 for clarification.

5. Line 55: Remove "until now," as it contradicts the rest of the sentence.

**Reply:** Thank you for your comment. We have removed it. The change is shown on page 3, line 65, in the marked-up manuscript version.

---

## Author Response (AR3)

NHESS-2023-192

**Authors' Responses to Reviewer 2 (RC2, anonymous)**

Date: 16 March 2025

Title: Investigation of an extreme rainfall event during 8–12 December 2018 over central Viet Nam – Part 2: An evaluation of predictability using a time-lagged cloud-resolving ensemble system

Authors: C. C. Wang et al.

Firstly, **we thank the reviewers for spending valuable time reviewing the paper again and giving us constructive comments that helps to improve the clarity of the paper.**

These are our responses to reviewers:

**Minor Comments (Comments are not listed in order of importance):**

1. Line 51: Should this be figure 1, as opposed to figure 4?

Reply: We apologize for causing the confusion. At line 51, we do mean Fig. 4 (OBS) because we referred to the observed accumulated rainfall of the rainiest day (10 December), not the 3-day rainfall.

2. Lines 98-99: Remove "has become indispensable for its ability to" and change "simulate" to simulates.

Reply: Thank you for your suggestion. We have removed the mentioned words and corrected the sentence, as you suggested. The change is shown on page 5, lines 84-85, in the marked-up manuscript version.

3. Lines 106-111: Yes, generally over long periods the ensemble mean is more accurate. However, it should be noted that it will smooth out the precipitation field and likely not capture the extreme magnitude of events.

Reply: Thank you for your comment. We totally agree that while the ensemble mean generally improves overall accuracy, it may smooth out extreme events.

Therefore, we have clarified this limitation in the revised version. The change is shown on page 5, lines 96-97, in the marked-up manuscript version.

4. Lines 142-155: There is a scale dependence to the practical and intrinsic predictability as well. Likely worth noting in this section.

Reply: Thank you for your suggestion. We also totally agree that predictability is scale-dependent and have included this information in the revised version. Moreover, we have added some references as supporting evidence. The change is shown on page 6, lines 125-126, in the marked-up manuscript version.

5. Lines 156-189: Need to break this into smaller paragraphs. I suggest one at the end of line 172 and 179.

Reply: Thank you for your suggestion. We have split the mentioned paragraph into smaller ones, as you suggested. The change is shown on page 7, lines 152 and 159, in the marked-up manuscript version.

6. Line 256: I would remove the word "retrieval" when talking about the TIGGE data. This leads to some confusion in my opinion, as "retrieval" has connotations of a mathematical processing; not just getting the data.

Reply: Thank you for your suggestion. We have removed it, as you suggested. The change is shown on page 10, lines 231 and 234, in the marked-up manuscript version.

7. Section 2.3.1: If I am understanding the SSS correctly, it does not take into account neighborhoods when verifying the precipitation? This would be something to specify, if so. Further, using a verification method that allows for a precipitation object to be slightly off location but still have a similar shape would be a useful comparison. Fraction skill score could be an option.

Reply: Thank you for your comments/suggestion. You are right that the SSS does not account for neighborhoods in precipitation verification. We have clarified this in the revision. The change is shown on page 15, lines 307-309, in the marked-up manuscript version.

Additionally, we appreciate your suggestion regarding alternative verification methods, such as the Fraction Skill Score. However, we prefer to keep our current method using SSS because it is suited for QPF verifications where the rainfall location is important (as in our case).

8. Lines 392: "This is true for a well calibrated ensemble, only." Does not seem to go here. I know this was a comment I had from the last round of revisions…but I am not sure why it got placed here.

Reply: Thank you for your comments. We sincerely apologize for misunderstanding your point. We have removed that sentence and revised the text in Section 2.3.2 to clarify your intended meaning. Besides, we also have added a reference as supporting evidence. The change is shown on page 15, lines 314-316, and page 16, line 352 in the marked-up manuscript version.